# A Concept-Centric Approach to Multi-Modality Learning

**Yuchong Geng**  *yg534@cornell.edu*
*School of Electrical and Computer Engineering*
*Cornell University*

**Ao Tang**  *atang@cornell.edu*
*School of Electrical and Computer Engineering*
*Cornell University*

**Reviewed on OpenReview:** *https:// openreview. net/ forum? id= 8WAAPP32c7*

## Abstract

Humans possess a remarkable ability to acquire knowledge efficiently and apply it across diverse modalities through a coherent and shared understanding of the world. Inspired by this cognitive capability, we introduce a concept-centric multi-modality learning framework built around a modality-agnostic concept space that captures structured, abstract knowledge, alongside a set of modality-specific projection models that map raw inputs onto this shared space. The concept space is decoupled from any specific modality and serves as a repository of universally applicable knowledge. Once learned, the knowledge embedded in the concept space enables more efficient adaptation to new modalities, as projection models can align with existing conceptual representations rather than learning from scratch. This efficiency is empirically validated in our experiments, where the proposed framework exhibits faster convergence compared to baseline models. In addition, the framework's modular design supports seamless integration of new modalities, since projection models are trained independently yet produce unified outputs within the shared concept space.

We evaluate the framework on two representative downstream tasks. While the focus is not on task-specific optimization, the framework attains comparable results with a smaller training footprint, no task-specific fine-tuning, and inference performed entirely within a shared space of learned concepts that offers interpretability. These findings point toward a promising direction for developing learning systems that operate in a manner more consistent with human cognitive processes.

## 1 Introduction

Humans are capable of acquiring knowledge at remarkable speed even from a young age, which stands in stark contrast to most learning frameworks that require substantial resources to achieve human-like intelligence on specific tasks. Moreover, human cognition is grounded in a shared and coherent understanding of the world that spans across different modalities. For instance, when learning a new language, we do not build an entirely separate system of knowledge for it. Instead, we intuitively connect new linguistic elements to our existing understanding of the world, or in other words, to our common sense. We believe a concept-centric approach to multi-modality learning could be key to not only bridging the efficiency gap, but also enabling learning frameworks that are more structured, reusable, and aligned with how humans organize knowledge across modalities.

At the center of our framework[1] is a concept space that carries universal knowledge applicable to diverse modalities, resembling the common sense embedded in the human mind. Recent inspiring works on Concept Learning often focus on linking concepts to specific neurons (Liu et al., 2023b) and encoded embedding

---

[1]Code is available at `https://github.com/Yuchong-Geng/concept-centric-multimodal-learning`.

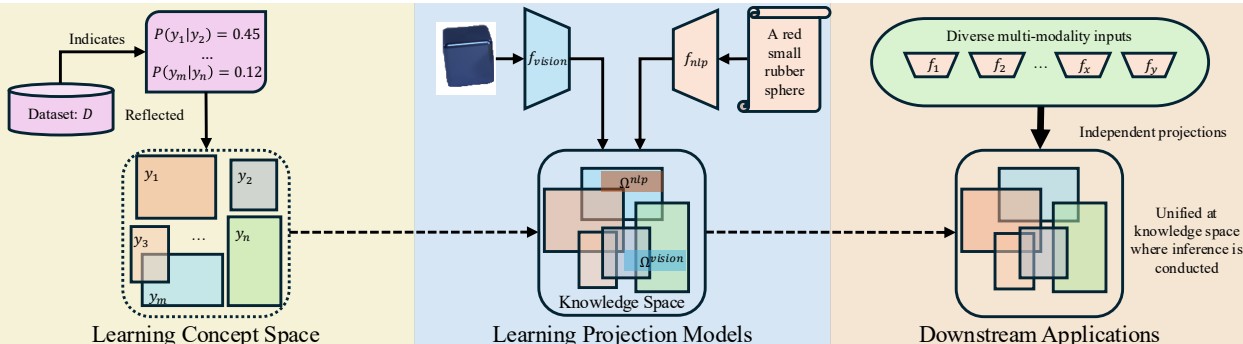

**Figure 1:** Overall structure of the proposed concept-centric multi-modality learning framework. A modality-agnostic concept space is trained to reflect the relations between the set of concepts $\mathcal{Y}$ as observed in a training dataset $\mathcal{D}$ (left). Modality-specific projection models are trained to create projections $\Omega$ for their inputs based on the inputs' associations with concepts (middle). The modular design of the framework offers great flexibility and adaptability to a wide range of downstream tasks (right).

vectors (Kalibhat et al., 2023; Wang et al., 2023b) of a model or injecting specific concepts as neurons into a model's structure (Sheth & Kahou, 2023; Koh et al., 2020). Compared to these works, our proposed framework takes a systematic approach by organizing modality-agnostic abstract concepts in an interpretable knowledge space and establishing connections to different modalities by projecting modality-specific inputs onto the same space.

While it is common in multi-modality learning to create a shared representation space for multiple modalities (Radford et al., 2021; Li et al., 2022; Ramesh et al., 2022) or utilize projections to align features from different modalities (Liu et al., 2023a; Hsiung et al., 2022), such approaches primarily aim to co-locate modality-specific features in a common embedding space. In contrast, our shared concept space serves as an explicit intermediate layer of abstract concepts that can be reused and composed across modalities, enabling more efficient learning and seamless incorporation of new modalities, as demonstrated in our experiments. We believe the proposed framework takes a step toward matching key aspects of human learning, where we build cohesive concept-level understanding and connect multiple modalities (e.g., vision and language) to shared knowledge.

Specifically, as outlined in Fig. 1, the proposed multi-modality learning framework features an abstract concept space and a set of modality-specific projection models. The modality-agnostic concept space, inspired by prior works on structured embedding spaces (Vilnis et al., 2018; Li et al., 2018), optimally reflects real-world relations between concepts via entailment probabilities (Fig. 1, left). Probing this concept space can also be achieved through simple queries of concept pairs of interest, providing interpretability into the learned knowledge.

Complementing the concept space, modality-specific projection models process inputs from different modalities and map them into a shared domain, which we refer to as the knowledge space (Fig. 1, middle). This knowledge space hosts both the abstract knowledge encoded in the concept space and the specific information extracted from individual inputs. By decoupling the projection models from the concept space, the framework enables efficient and modular learning. Each projection model is only required to produce consistent outputs within the knowledge space, allowing flexibility in architecture and optimization for different modalities. Although the projection models operate independently, their outputs are unified in the knowledge space, where they can interact with each other and with the learned concept representations, resulting in a structure that supports probabilistic reasoning and cross-modality interactions.

The proposed design, characterized by a shared concept space with universally applicable knowledge and flexible projection mechanisms, naturally facilitates the reuse of learned knowledge across diverse modalities and task domains. Such a design enhances the generalizability of our framework and enables straightforward adaptation to various downstream tasks, with all inference processes conducted within the knowledge space (Fig. 1, right).

**Contribution.** Our contributions are three-fold. *First*, we propose a novel approach to multi-modality learning that centers around a concept space embedded with universally applicable knowledge. To our knowledge, this idea of a concept-focused learning scheme has been underexplored in the field of multi-modality learning (Sec. 3). *Second*, we offer a clear motivation and justification for the proposed framework. Leveraging knowledge learned from the concept space, our framework demonstrates more efficient learning curves compared to traditional methods (Fig. 2). Additionally, although our projection models are trained independently, they still exhibit strong cross-modality alignment without joint training because they rely on the same concept space that encodes shared abstract knowledge. (Sec. 4.2) This design also makes the framework naturally scalable for incorporating new and diverse modalities (Sec. 4.3). *Third*, we evaluate the performance of our framework on two downstream tasks. We show that the proposed framework, with a modest pretraining footprint, achieves comparable performance to benchmarks out-of-the-box without fine-tuning, while conducting all inference within a shared knowledge space containing interpretable concept representations (Sec. 4.4).

## 2 Related Work

**Multi-Modality Learning.** Vision and language modalities remain at the forefront of multi-modality learning research, with some works exploring alternative modalities like audio (Akbari et al., 2021; Shi et al., 2022) and biomedical data (Masood et al., 2025). Within the vision-language area, CLIP by Radford et al. (2021) employs two modality-specific encoders to learn a joint representation through image-text matching. Subsequent work by Ramesh et al. (2022) introduces a text-to-image generation framework, using a text encoder and an image decoder to generate high-quality images from textual descriptions. Transformer-based architectures (Vaswani et al., 2017) have been widely explored for cross-modality information exchange and learning (Singh et al., 2022a; Bao et al., 2022; Kim et al., 2021a; Wang et al., 2023a; Lu et al., 2023).

Beyond combining and relating modalities, research has delved into diverse areas such as multi-modality few-shot learning (Alayrac et al., 2022; Li et al., 2021) and visual-textual pattern mining (He & Peng, 2020). Some studies propose generalized learning frameworks applicable across various modalities (Jaegle et al., 2021; Baevski et al., 2022a;b). While these frameworks showcase strong capabilities in tasks like text-to-image generation and visual-language few-shot learning, our work addresses a distinct and important issue: creating a universally applicable concept space with abstract knowledge reflecting real-world observations. Baevski et al. (2022b) present a versatile representation learning framework, yet it isolates modalities, impeding cross-modality interactions. In contrast, our proposed method directly combines modalities by projecting modality-specific inputs onto a unified concept space, eliminating the information barrier between them.

**Concept Learning.** Early approaches to Concept Learning utilized Boolean logic for defining concepts based on relationships with other concepts (Angluin, 1988) and their associated attributes (Mitchell, 1997). Lake et al. (2015) propose a Bayesian Program Learning framework that represents concepts as probabilistic programs. Nowadays, a prevalent method involves placing concepts within a structured embedding space. Concept learning frameworks such as those proposed by Mao et al. (2019) and Li et al. (2020b) construct embedding spaces that align concept representations with corresponding visual feature vectors. Lee et al. (2024) propose a framework that learns concept embeddings via distillation from pre-trained vision-language models. Methods from Vilnis et al. (2018) and Mei et al. (2022) emphasize entailment relationships between concepts in learned embedding spaces, while the work from Sinha et al. (2024) captures hierarchical information.

In a departure from structured concept embedding spaces, the Concept Bottleneck Model (CBM) (Koh et al., 2020) has become a popular framework that represents concepts as intermediate neural network outputs. CBM first predicts a set of pre-defined concepts aligned with human annotations and then produces a classification decision based on those concept predictions. Extending the idea of CBM, Dominici et al. (2023) propose a framework that learns a multimodal concept space jointly across modalities, using CBM-style supervision to inject interpretability into the shared concept space. Liu et al. (2023b) propose a method for identifying a small subset of model parameters responsible for generating specific concepts in a diffusion model. Kong et al. (2024) propose a theoretical view of concept learning as an identification problem of a discrete latent hierarchical model.

While we acknowledge that some motivating works adopt a similar strategy involving a concept embedding space, our approach stands out for several reasons. The primary distinction lies in the organization of our concept space, which reflects real-world knowledge by providing meaningful numerical entailment probabilities that mirror relationships among actual concepts. Furthermore, no barrier in our concept space prevents concepts belonging to different groups, such as *red* in `color` and *cube* in `shape`, from interacting with each other. More importantly, instead of being fitted to a specific modality, our concept space is designed to be abstract and modality-agnostic, thus allowing interactions between inputs from different modalities.

## 3 Method

Our proposed multi-modality learning framework consists of a modality-agnostic concept embedding space that models underlying relationships between concepts via entailment probabilities and a set of modality-specific projection models that extract representation from single-modality inputs and project them onto the domain where the concept space resides, i.e., the knowledge space.

Learning abstract knowledge in the concept space ensures generality, which makes its domain a good landing place for extracted representations from different modalities. Decoupled from the concept space and each other, modality-specific projection models can be tailored for adaptation to their unique inputs, while modality-specific knowledge remains connected after the projection.

We describe the design of the concept space in Sec. 3.1 and projection models in Sec. 3.2. Further implementation details can be found in Sec. 4.1.

### 3.1 Learning Concept Space

Davis et al. (1993) describe a knowledge representation as a surrogate that both carries the *thing* existing in the real world and serves as a medium for pragmatically efficient computation. Building upon their definition of knowledge representation, we adopt a probabilistic box embedding space (Li et al., 2018) to organize the learned representations of abstract concepts. In this formulation, each concept is represented as an axis-aligned hyperrectangle (box) in a high-dimensional space, and geometric relations between boxes correspond directly to semantic relations. Intuitively, the strength of the relationship between two concepts is reflected in the amount of overlap between their boxes: a larger intersection indicates stronger semantic inclusion or entailment, while boxes with smaller overlap correspond to weaker or less related concepts.

Like mental entities of specific knowledge in our brains, where we can relate concepts to each other, abstract entities in this concept space should be capable of interacting with each other, allowing reasoning inferences. With probabilistic box embeddings, these interactions take a geometric form: relations between concepts are expressed through the overlap structure of their boxes, which determines their entailment probabilities. In the proposed framework, we focus on entailment relations between concepts depicted by these entailment probabilities to allow interactions between concepts. Contrary to latent spaces or learned ML model parameters, probing into the learned knowledge of this concept space can be easily achieved by querying the entailment probabilities of concept pairs of interest. Furthermore, our experiments demonstrate the efficiency of learning and referencing this concept space, facilitated by its compact parameter size, which qualifies it as *a medium for pragmatically efficient computation.*

**Defining Concept Space.** We first define a knowledge space $\mathcal{K} \subset \mathbb{R}^d$ as a $d$-dimensional embedding space. Let $\mathcal{Y}$ be a set for modality-agnostic concepts. Each concept $y \in \mathcal{Y}$ is represented in $\mathcal{K}$ by a box embedding (the *surrogate*), defined by a pair of vectors $\Omega_y = (\omega_{\min,y}, \omega_{\max,y})$, where $\omega_{\min,y}, \omega_{\max,y} \in \mathcal{K}$ correspond to the minimum and maximum boundaries of the box in $\mathcal{K}$. We use $\mathcal{C} = \{\Omega_y \mid y \in \mathcal{Y}\} \subset \mathcal{K}$ to denote a set of box embeddings for every concepts in $\mathcal{Y}$ and we call $\mathcal{C}$ the concept space whose parameters are optimized to reflect real-world knowledge.

A smoothing function $m_{\text{soft}}^i(\omega) = \frac{\text{softplus}(\omega^i)}{\text{softplus}(G_{\max}^i - G_{\min}^i)}$ is introduced on each dimension $i$ of $\mathcal{K}$ so a joint probability between two disjoint concepts can still be obtained. $G_{\max}^i, G_{\min}^i$ terms are the global maximum and minimum values at the $i$ dimension among all $\Omega_y$s in $\mathcal{C}$. More details of $m_{\text{soft}}^i$ can be found in

Appendix A.1. The probability of a single concept $y$ is calculated as $P(y) = P(\Omega_y) = \prod_{i=1}^{d} m_{\text{soft}}^{i}(\omega_{\max,y} - \omega_{\min,y})$. The joint probability between two concepts $y_1$ and $y_2$ is calculated as

$$P(y_1 \cap y_2) = P(\Omega_{y_1} \cap \Omega_{y_2}) = \prod_{i=1}^{d} m_{\text{soft}}^{i}(\min(\omega_{\max,y_1}, \omega_{\max,y_2}) - \max(\omega_{\min,y_1}, \omega_{\min,y_2}))$$

**Embedding Knowledge.** Let $\mathcal{X}_*$ denote a sample space of an unspecified modality marked by *, where each sample can be associated by a subset of modality-agnostic concepts in $\mathcal{Y}$. A training dataset is given as $\mathcal{D}_* = \{(x_i^*, \boldsymbol{y}_i)\}_{i=1}^{N}$, where $x_i^* \in \mathcal{X}_*$ and $\boldsymbol{y}_i = \{y_j \mid y_j \in \mathcal{Y} \text{ and } y_j \text{ describes } x_i^*\}$. This set of concepts that describe $x_i^*$ can include both attribute concepts, like *fluffy* and *blue*, as well as category concepts, like *dog* and *sky*.

Modality-agnostic abstract knowledge can be extracted from $\mathcal{D}_*$ by examining entailment probabilities between concepts indicated by $\{\boldsymbol{y}_i\}_{i=1}^{N}$. Specifically, the ground-truth probability of a single concept and the entailment probability of a concept pair $(y_1, y_2)$ are calculated by $P(y) = \frac{\text{count}(y)}{\sum_{y' \in \mathcal{Y}} \text{count}(y')}$ and $P(y_1 \mid y_2) = \frac{\text{count}((y_1 \cap y_2))}{\text{count}(y_2)}$ as they appear in $\mathcal{D}_*$.

To drive the concept space to reflect real-world relationships between concepts via entailment probabilities, the objective for pretraining $\mathcal{C}$ is naturally defined as minimizing the KL divergence between predicted probabilities obtained from $\mathcal{C}$ and the true probabilities observed in $\mathcal{D}_*$. In addition to the true concepts in $\boldsymbol{y}_i$ for each data point, we sample a set of negative concepts and append them to $\boldsymbol{y}_i$. This negative sampling is necessary because the supervision available in $\mathcal{D}_*$ contains only positive co-occurring concept pairs; without explicit counterexamples, the model would have no evidence that unrelated concepts should exhibit low or zero entailment. Negative samples therefore provide the missing contrastive signal, ensuring that the concept space learns to represent both strong entailments (via overlapping boxes) and non-entailments (via small or negligible overlap), rather than collapsing into an overly permissive representation. The specific sampling protocol varies across datasets, and further details are provided in Sec. 4.

For each sample and its set of concept labels $(x_i^*, \boldsymbol{y}_i) \in \mathcal{D}_*$, we compute the entailment probability $Q(y_1 \mid y_2)$ given by the concept space for every possible combination of concept pairs $(y_1, y_2)$ and for sampled negative concept pairs in $\boldsymbol{y}_i$, and we compare these values with the corresponding true entailment probabilities $P(y_1 \mid y_2)$. We refer readers to Appendix A for more details about the concept space.

## 3.2 Learning Projection Models

**Defining Projection Models.** Decoupled from the abstract concept space, each modality-specific projection model can be viewed as a mapping function $f_* : \mathcal{X}_* \to \mathcal{K}$ that generates a box representation in $\mathcal{K}$ for each input from its modality-specific sample space $\mathcal{X}_*$ of an unspecified modality denoted by *. This projection onto $\mathcal{K}$ allows interactions between specific objects from $\mathcal{X}_*$ and abstract concepts in $\mathcal{C}$. Specifically, given a modality-specific input $x_i^* \in \mathcal{X}_*$, its representation in $\mathcal{K}$ can be obtained by $f_*(x_i^*; \theta) = \Omega_i^*$ where $\Omega_i^* \subset \mathcal{K}$ follows the same definition of $\Omega_y \subset \mathcal{C}$. With this representation made available, the probability that an object is associated with a concept $y$ can be naturally described by an entailment probability of $P(y \mid x_i^*) = P(\Omega_y \mid \Omega_i^*)$.

**Adapting to the Concept Space.** Given the training set $\mathcal{D}_*$ corresponding to a modality marked by *, the projection produced for an input $x_i^*$ should entail not only a single concept $y$ but also **all** other concepts associated with $x_i^*$. In other words, the projection $\Omega_i^*$ for $x_i^*$ should lie at the **intersection** of the set of concepts describing $x_i^*$. Thus, the optimal projection for $x_i^*$ should maximize the entailment probability $P(\bigcap_{y_j \in \boldsymbol{y}_i} y_j \mid x_i^*)$.

In our experimental datasets, concepts fall into two types: attribute concepts and category concepts. A sample may possess *multiple* positive attributes but is associated with *exactly one* mutually exclusive category. As a result, attribute concepts require independent binary classification, whereas category concepts require a single multi-class decision. Naturally, a binary cross-entropy loss is applied to attribute concepts,

and a SoftMax cross-entropy loss is applied to category concepts. Further details about the datasets and the nature of these concept types are provided in Sec. 4.

To drive projection models to produce the optimal projection described above, we use a combination of a binary cross-entropy loss on attribute concepts $\mathcal{Y}^{\text{attr}} \subset \mathcal{Y}$:

$$
\begin{aligned}
\ell_{\text{attr}}(\boldsymbol{y}, \Omega_*) = \frac{1}{|\mathcal{Y}^{\text{attr}}|} \sum_{y \in \mathcal{Y}^{\text{attr}}} & \mathbb{I}(y \in \boldsymbol{y})[-w \cdot \log P(\Omega_y \mid \Omega_*)] \\
& + \mathbb{I}(y \notin \boldsymbol{y})[\log(1 - P(\Omega_y \mid \Omega_*))]
\end{aligned}
\tag{1}
$$

(where $w$ is a weight assigned to positive attribute concepts)

and a multi-class cross-entropy loss with SoftMax on category concepts $\mathcal{Y}^{\text{cat}} \subset \mathcal{Y}$:

$$
\ell_{\text{cat}}(\boldsymbol{y}, \Omega_*) = -\log \frac{\exp P(\Omega_{y^{\text{cat}}} \mid \Omega_*)}{\sum_{y' \in \mathcal{Y}^{\text{cat}}} \exp P(\Omega_{y'} \mid \Omega_*)}
\tag{2}
$$

(where $y^{\text{cat}} \in \boldsymbol{y}$)

Now, given a specific modality denoted by A and its training dataset $\mathcal{D}_A$. The training objective for $f_A$ is formally described as minimizing:

$$
\mathcal{L}_A(\theta_A; \mathcal{D}_A) = \frac{1}{|\mathcal{D}_A|} \sum_{(x, \boldsymbol{y}) \in \mathcal{D}_A} \ell_{\text{attr}}(\boldsymbol{y}, f_A(x; \theta_A)) + \ell_{\text{cat}}(\boldsymbol{y}, f_A(x; \theta_A))
\tag{3}
$$

While the training objective and projection outputs remain consistent across different modalities, projection models can be customized to accommodate unique modality-specific inputs, such as images or sequences of texts, bringing flexibility and versatility to the proposed framework.

### 3.3 Adapting to Downstream Tasks

With an abstract concept space and decoupled projection models, our proposed learning framework naturally accommodates various downstream tasks involving single or multiple modalities. Regardless of the specific downstream tasks, their inference process consists of two stages: creating projections and relating them to learned knowledge. This approach more closely resembles human learning than traditional black-box models. In our daily interactions with objects, we process external stimuli like vision by creating abstract mental entities for objects we see. We then comprehend these mental entities using our understanding of the world, or, in other words, our concept space (Gärdenfors, 2014). In Section 4, we use an Image-Text Matching task involving multi-modality and a Visual Question Answering task with a single-modality-focused approach to illustrate the functionality of the proposed framework.

## 4 Implementation and Experiments

We base our evaluation on three existing datasets: CLEVR (Johnson et al., 2017a), COCO (Lin et al., 2014), and GQA (Hudson & Manning, 2019), where their concepts are formed from original and supplemental annotations (Patterson & Hays, 2016). COCO and GQA contain both attribute and category concepts, while CLEVR provides only attribute concepts. For consistency across datasets, the `shape` attribute family in CLEVR is reinterpreted as category concepts when required.

In COCO and GQA, many annotated attributes appear extremely infrequently (sometimes only a few dozen instances), which provides insufficient training signal. To ensure meaningful supervision, we focus on the most frequent and well-represented concepts, retaining the top 35 attribute concepts for COCO and an

equally sized set of 35 attribute concepts for GQA, together with their associated categories. This selection yields 64 total concepts for COCO and 68 for GQA.

To evaluate the scalability and generalization capability of our framework, we also construct a Unified dataset that aggregates all concepts and modality-specific representations from CLEVR, COCO, and GQA. This dataset contains overlapping concepts across domains, enabling us to examine how well the framework transfers and reconciles abstract concepts when they appear in different visual or linguistic forms. More details on the datasets and preprocessing steps can be found in Appendix C. Our experiments follow the same train and validation splits as the original datasets: the framework is pretrained on the train sets and evaluated on the validation sets.

## 4.1 Pretraining

**Concept Space.** To ensure that each concept box always has a valid set of lower boundaries smaller than its upper boundaries, we use two vectors, $(\omega_{\min,y}, \omega_{\Delta,y}) = \Omega_y$, instead of $(\omega_{\min}, \omega_{\max})$ to represent a box in our actual experiments, where $\omega_\Delta \in \mathcal{K}_{\geq 0}$ is restricted to non-negative values. A box's upper boundaries can be obtained by $\omega_{\max} = \omega_{\min} + \omega_\Delta$. We set the dimension of $\mathcal{K}$ to 50, based on empirical experiments (see Appendix A.5 for an ablation study on concept space dimensionality). Initial parameters for $\mathcal{C}$ are sampled from two uniform distributions. As for the negative sampling method, in CLEVR, the only negative concept pairs come from combinations of concepts residing in the same-attribute families, such as (*red*, *blue*) in the `color` family. For COCO and GQA, negative samples are randomly selected from all concepts. The concept space is trained for just two epochs for each dataset with a batch size of 256 using an AdamW optimizer (Loshchilov & Hutter, 2017) with a learning rate of $10^{-3}$. The training of this concept space can be completed quickly as there are only thousands of parameters for a moderately-sized concept space.

**Projection Models.** In adapting our framework to the datasets featuring vision and natural language modalities, we incorporate a vision projection model $f_{\text{vision}}$ based on a Vision Transformer encoder (Dosovitskiy et al., 2020) and a natural language projection model $f_{\text{NL}}$ based on a BERT encoder (Devlin et al., 2018). Both models utilize their encoders' outputs on `[CLS]` tokens to generate projection boxes in $\mathcal{K}$. The outputs $e$ with a dimension of 768 are divided into two equal chunks, $h_{\min}$ and $h_\Delta$, each with a dimension of 384. These chunks are then input into two fully connected layers to produce $\omega_{\min}$ and $\omega_\Delta$ for their respective projection boxes. To ensure $\omega_\Delta$ is always a non-negative vector, an additional ReLU layer is applied. The complete projection process for inputs from the vision modality is outlined in Algorithm 1.

---

**Algorithm 1** Illustration of a ViT-based projection model $f_{\text{vision}}$ which projects vision modality inputs to the knowledge space $\mathcal{K}$

---

**input** modality-specific input $x_{\text{vision}}$

**Ensure:** $\omega_\Delta^{\text{vision}} \in \mathcal{K}_{\geq 0}, \Omega^{\text{vision}} \subset \mathcal{K}$

$\quad e_{\text{vision}} \leftarrow \text{ViT}(x_{\text{vision}})$

$\quad h_{\min}^{\text{vision}}, h_\Delta^{\text{vision}} \leftarrow \text{split}(e_{\text{vision}})$

$\quad \omega_{\min}^{\text{vision}} \leftarrow \text{Linear}_{\min}^{\text{vision}}(h_{\min})$

$\quad \omega_\Delta^{\text{vision}} \leftarrow \text{ReLU}(\text{Linear}_\Delta^{\text{vision}}(h_\Delta))$

**output** $\Omega^{\text{vision}} = (\omega_{\min}^{\text{vision}}, \omega_\Delta^{\text{vision}})$

---

For each object $i$ in the CLEVR dataset, its attribute prediction for a specific attribute family $\boldsymbol{z}$ (e.g., `color`) is generated by $\bar{y}_i^z = \text{argmax}_{y \in \boldsymbol{z}} P(\Omega_y | \Omega_i)$. For each object $i$ in COCO and GQA, a threshold is applied to $P(\Omega_y \mid \Omega_i), y \in \mathcal{Y}^{\text{attr}}$ to obtain attribute predictions, and category prediction is generated by $\bar{y}_i^{\text{cat}} = \text{argmax}_{y \in \mathcal{Y}^{\text{cat}}} P(\Omega_y \mid \Omega_i)$.

We establish a baseline by replacing the concept space with a traditional Multilayer Perceptron (MLP) at the classification head of $f_{\text{vision}}$. Additionally, we implement the vision-modality projection model using a ResNet model (He et al., 2015) as the backbone to showcase the flexibility of the proposed framework.

Results summarized in Table 1 show that our proposed framework achieves comparable performance to traditional models while leveraging a novel concept space with interpretable learned knowledge.

| Backbone | Method | CLEVR | COCO | | GQA | | Unified | |
|---|---|---|---|---|---|---|---|---|
| | | Acc | Acc | mAP | Acc | mAP | Acc | mAP |
| ResNet | Baseline | 0.997 | 0.894 | 0.532 | 0.725 | 0.202 | 0.897 | 0.447 |
| | $f_{\text{vision}}$ | 0.990 | 0.892 | 0.520 | 0.727 | 0.191 | 0.892 | 0.439 |
| ViT | Baseline | 0.999 | 0.960 | 0.594 | 0.842 | 0.335 | 0.949 | 0.512 |
| | $f_{\text{vision}}$ | 0.999 | 0.957 | 0.590 | 0.844 | 0.352 | 0.949 | 0.502 |

**Table 1:** Comparison with baseline models on vision-modality classification. CLEVR uses Accuracy. COCO, GQA, and Unified evaluate category concepts using Accuracy and attribute concepts using mean Average Precision (mAP).

Apart from featuring a concept-centric learning scheme, the proposed framework can also learn modality-specific knowledge faster by referencing learned knowledge from the modality-agnostic concept space as indicated in Fig. 2. This more natural learning process of our framework bridges the efficiency gap between traditional machine learning methods, which often demand extensive data, and human learning, which excels at adeptly and efficiently extracting modality-specific representations and associating them with mental entities of abstract knowledge. To fully evaluate the impact of this transparent, modality-agnostic concept space on the learning of modality-specific projection models, we conduct an ablation study on it in Sec. A.4.

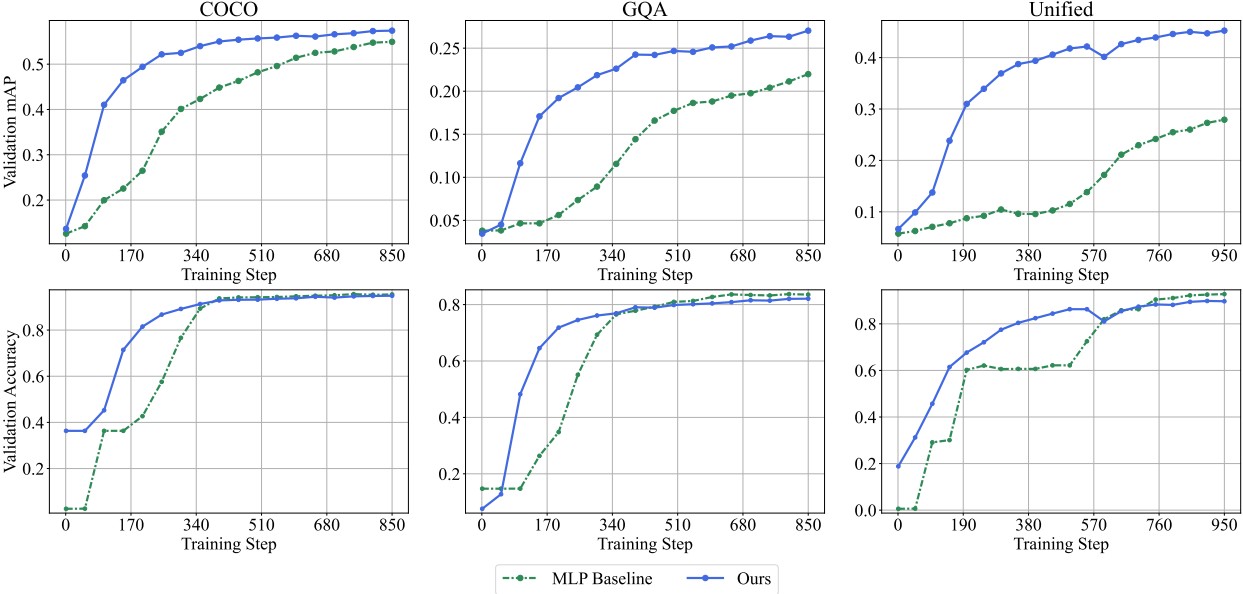

**Figure 2:** Learning curves of the proposed projection models and the baseline model. The plot displays the evaluation accuracy on category concepts and the evaluation mAP on attribute concepts measured every 50 training steps. During the learning process, the proposed vision-modality projection model improves more quickly compared to the baseline thanks to the universal concept space that already has abstract knowledge embedded in it. This faster learning process of our framework bridges the efficiency gap between traditional machine learning methods, which require a large amount of data, and human learning that excels at extracting modality-specific representations and linking them to structured abstract knowledge.

Projection models for the natural-language modality achieve highly accurate performance ($\geq 99\%$) thanks to the clearly structured description sentences. Further implementation and training details of projection models can be found in Appendix D.

| Model | Init. Acc. | Best Acc. | Time to 95% Acc. (s) |
|-------|-----------|-----------|----------------------|
| CLIP  | 0.767     | 0.9701    | 42.7                 |
| FLAVA | 0.499     | 0.9848    | 103.1                |
| ViLT  | 0.747     | 0.9568    | 265.1                |
| Ours  | 0.954     | 0.9568    | – (already >95% at step 1) |

**Table 2:** Multimodal alignment comparison across vision-language models. We use image-text matching accuracy for validation split of the Unified dataset as a proxy for evaluating cross-modality alignment. Our framework achieves high alignment at initialization (over 95% accuracy at step 1) due to the shared knowledge, and requires significantly less computation than CLIP, FLAVA, or ViLT to reach their best performance.

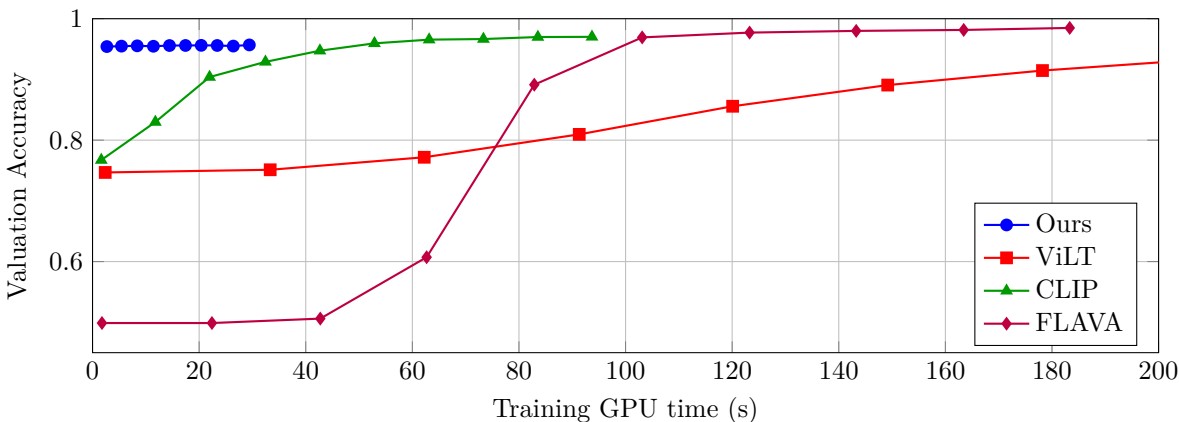

**Figure 3:** Comparison of training efficiency for cross-modality alignment. Validation accuracy for image-text matching task of the Unified dataset is used as a proxy for evaluating cross-modality alignment. Our proposed method reaches near-saturated alignment without finetuning and attains its final accuracy within a significantly shorter period, while ViLT, CLIP, and FLAVA require substantially more computation time to achieve comparable alignment.

## 4.2 Cross Modality Alignment

A key distinction between our framework and most existing multimodal models is that each modality-specific projection model is trained independently, yet they remain inherently compatible and aligned because they share a common concept space that already encodes abstract knowledge. As a result, the projections from different modalities are well aligned without requiring any extensive joint training typically needed in models such as ViLT, CLIP, or FLAVA. We assess alignment between the vision and language modalities from our decoupled projection models by measuring the agreement between their knowledge space projections and by comparing the results with strong multimodal benchmark models. Because cross-modality alignment cannot be measured directly, we use accuracy on an image–text matching task (Section 4.4.1) from the Unified dataset with a large pool of 130 concepts as a proxy, and we initialize all comparison models with their pretrained weights to ensure fairness.

Before any fine-tuning, we measure each method's initial alignment. Table 2 shows that our independently trained projection models already achieve strong cross-modal alignment at initialization, reaching over 95% accuracy without further training. We then apply a joint training objective from Appendix B to finetune the two projection models. This joint training is used only to obtain a direct comparison of cross-modality alignment efficiency against existing multimodal models and is *not required* for our framework to function. Figure 3 plots image–text matching accuracy every 50 steps against the GPU time elapsed during finetuning. Our framework not only starts with strong alignment but also converges significantly more quickly than existing multimodal models during joint training.

| Modality | Image | English | Chinese | Spanish | French |
|----------|-------|---------|---------|---------|--------|
| **Image** | – | 0.955 | 0.968 | 0.961 | 0.935 |
| **English** | 0.955 | – | 0.974 | 0.982 | 0.876 |
| **Chinese** | 0.968 | 0.974 | – | 0.970 | 0.887 |
| **Spanish** | 0.961 | 0.982 | 0.970 | – | 0.878 |
| **French** | 0.935 | 0.876 | 0.887 | 0.878 | – |

**Table 3:** Cross-modality matching accuracy across image, English, Chinese, Spanish, and French. Each modality-specific model is trained independently with no joint training.

### 4.3 Incorporating New Modality

Thanks to the decoupled training scheme of each individual modality-specific projection model and the inherent compatibility and alignment that result from sharing a unified knowledge space, incorporating new modalities into our framework is easy and scalable. To add a new modality to a functioning system with existing projection models, the new projection model only needs to be trained using the same projection-model training objective, and none of the existing modality-specific models need to be adjusted or jointly finetuned.

To demonstrate this scalable design, we translated the natural language descriptions originally used to train the BERT-based projection model into three new languages, including Chinese, French, and Spanish. Each translated language represents a new modality with a distinct input space added to a system that previously contained only vision and English (previously referred to as the natural language modality). Although all inputs are natural language, each language is treated as an independent modality because it is mapped into the shared knowledge space via its separately trained modality-specific projection model, rather than sharing parameters with the English text projection as in multilingual encoders. We follow the same training process as for the English projection model and train these new models independently while adapting them to the shared knowledge space. We then directly measure cross-modality alignment using a cross-modality representation-pair matching task made from the Unified dataset as a proxy. Results in Table 3 show strong alignment across all modalities. To gain further insight into the agreement between different projection models, Figure 4 reports the average entailment probabilities for all positive cross-modality representation pairs (two representations correspond to the same set of concepts) and negative ones (perturbed concepts) across all modality combinations.

This ability to incorporate new modalities without any joint finetuning with existing modalities highlights the flexibility and scalability of our framework. In addition, the modality-specific projection models for these new modalities demonstrate the same efficient training behavior as shown in Figure 2.

### 4.4 Downstream Tasks

Now, we focus on our proposed framework's adaptation to two downstream tasks: Image-Text Matching involving cross-modality references and Visual Question Answering with a single-modality-focused approach.

#### 4.4.1 Image-Text Matching

Image-text matching is a binary classification task on whether a natural language sentence describes an image. Our framework can naturally adopt a common approach involving creating representations for sentences and images in a shared latent space. In contrast to those works, however, our latent space is a knowledge-embedded concept space that supports efficient probing. Specifically, given an image-text pair $(x_m^{\text{vision}}, x_n^{\text{NL}})$, their representations in the learned concept space $\mathcal{C}$ are generated by $f_{\text{vision}}(x_m^{\text{vision}}) = \Omega_m^{\text{vision}}$ and $f_{\text{NL}}(x_n^{\text{NL}}) = \Omega_n^{\text{NL}}$. The probability that $(x_m^{\text{vision}}, x_n^{\text{NL}})$ is a positive pair can be determined by the cross entailment probability as follows:

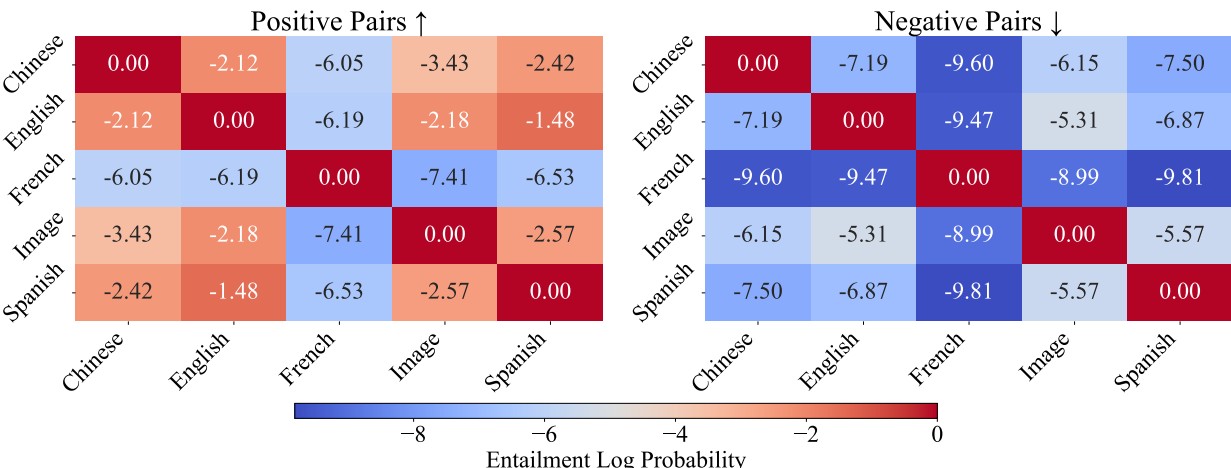

**Figure 4:** Modality alignment measured by entailment probabilities between positive and negative cross-modality representation pairs across image, English, Chinese, Spanish, and French modalities. Left: higher values (↑) indicate stronger projection overlap for positive concept pairs. Right: lower values (↓) indicate stronger separation for negative pairs. All modality-specific models are trained independently.

$$P(\text{matched} \mid (x_m^{\text{vision}}, x_n^{\text{NL}})) = \frac{1}{2} \left[ P(\Omega_m^{\text{vision}} \mid \Omega_n^{\text{NL}}) + P(\Omega_n^{\text{NL}} \mid \Omega_m^{\text{vision}}) \right]$$

This inference process is demonstrated in Fig. 8 in Appendix.

In our experiments, we employ two methods to create negative image-text pairs: swapping whole description sentences and swapping attributes. Specifically, for the first method, we replace 50% of images' description sentences using random sampling. For example, an original description sentence of a CLEVR object might be changed from "There is a large, metal, red cube" to "There is a *rubber, small, yellow sphere.*" On the other hand, swapping attributes involves changing only a subset of attributes that describe an object, creating a more challenging image-text matching task. For instance, the same description sentence would be changed to "There is a *small,* metal, red cube."

To compare our framework's performance, we implement other benchmark multi-modality models with applications in the Image-Text Matching task. The results are summarized in Table 4. In contrast to those models with traditional black-box architectures, our framework displays a more efficient learning process and adopts a more transparent inference process without sacrificing its performance. Details of this experiment can be found in Appendix E.

| Method | Fine-tuned? | CLEVR | | COCO | | GQA | |
|---|---|---|---|---|---|---|---|
| | | sent. | attr. | sent. | attr. | sent. | attr. |
| BLIP (Li et al.) | ✓ | 0.999 | 0.999 | 0.992 | 0.536 | 0.979 | 0.576 |
| CLIP (Radford et al.) | ✗ | 0.997 | 0.997 | 0.974 | 0.587 | 0.945 | 0.532 |
| FLAVA (Singh et al.) | ✓ | 0.998 | 0.998 | 0.992 | 0.505 | 0.980 | 0.536 |
| ViLT (Kim et al.) | ✓ | 0.994 | 0.994 | 0.985 | 0.515 | 0.965 | 0.555 |
| Ours | ✗ | 0.995 | 0.995 | 0.970 | 0.550 | 0.924 | 0.531 |

**Table 4:** A comparison with state-of-the-art multi-modality models on the Image-Text Matching Task. We test these models and our framework using two variants of the matching task: swapping whole sentences (sents.) and swapping attributes (attr.). Classification accuracy (%) is reported.

### 4.4.2 Visual Question Answering

Visual Question Answering (VQA) evaluates an AI system's ability to reason about images by answering questions related to those images in a natural language format. For this task, we focus on the CLEVR dataset, whose questions are designed to include attribute identification, counting, comparison, spatial relations, and logical operations. Recently, several works (Johnson et al., 2017b; Yi et al., 2018; Mao et al., 2019; Li et al., 2020a; Mei et al., 2022) have focused on a neural-symbolic reasoning approach, using chains of symbolic programs to predict answers to these questions. Our framework's adaptation to VQA involves using a similar set of symbolic programs, but these programs operate on the knowledge space $\mathcal{K}$ containing interpretable concepts in $\mathcal{C}$ instead of the high-dimensional latent spaces used by previous works.

**Problem Formulation.** Given an image-question pair $\{X_i^{\mathrm{vision}}, q_i\}$ where $X_i^{\mathrm{vision}}$ is an original CLEVR image as shown in Fig. 7 and $q_i$ is a natural language question such as *"Are there more cubes than yellow things?"*, an AI system needs to generate an answer $o_i$ in the natural language format such as *"Yes"*.

**Symbolic Programs.** We design our symbolic programs as deterministic functions operating on $\mathcal{K}$. Precisely, we follow the same program definitions as proposed by Johnson et al. (2017a).

**Program Generator.** An LSTM model $\pi$ is used to process questions into sequences of programs: $\hat{z}_i = \pi(q_i)$. We follow the same pretraining procedure used in (Johnson et al., 2017b) to train this program generator. However, as there is no fine-tuning stage in our adaptation, the parameters in $\pi$ are frozen once pretraining is finished.

**Object Detection and Projection.** Similar to our pretraining process, we use $f_{\mathrm{detection}}$ to obtain a set of single-object images $\boldsymbol{x}_i^{\mathrm{vision}}$ from $X_i^{\mathrm{vision}}$ which are then fed into $f_{\mathrm{vision}}$ so their projections can be obtained. Additionally, each single object's coordinates predicted by $f_{\mathrm{detection}}$ are attached to its projection box so questions involving spatial relations can be inferred.

**Inference Process.** A correctly predicted program sequence $\hat{z}_i$ starts with a `Scene` function that returns all objects in an image and ends with a program that outputs the answer $o_i$. Intermediate programs take the output from previous programs as inputs, which is a recurring process until the final function. Our concept space $\mathcal{C}$ is mainly involved in attribute identification, following the same procedure used when evaluating projection models in Sec. 4.1. Spatial information for each object is obtained from the bounding-box coordinates predicted by $f_{\mathrm{detection}}$ and is passed to the corresponding symbolic functions that handle spatial reasoning. The complete inference process is also demonstrated in Fig. 9 in Appendix.

| Method | Accuracy | Fine-tuned? |
|---|---|---|
| SA+MLP (Johnson et al.) | 73.2 | ✓ |
| Dependency Tree (Cao et al.) | 89.3 | ✓ |
| Human (Johnson et al.) | 92.6 | N/A |
| RN (Santoro et al.) | 95.5 | ✓ |
| IEP (Johnson et al.) | 96.9 | ✓ |
| MDETR (Kamath et al.) | 99.7 | ✓ |
| NS-VQA (Yi et al.) | 99.8 | ✓ |
| Ours | 96.5 | ✗ |

**Table 5:** A comparison between our framework's performance and state-of-the-art models.

**Results.** We perform no fine-tuning on the concept space $\mathcal{C}$ and vision-modality projection model $f_{\mathrm{vision}}$ for the VQA task. A comparison to benchmark models summarized in Table 5 shows our framework achieves performance levels on par with those fine-tuned benchmark models.

## 5 Discussion

**A Cognition-Inspired Learning Paradigm.** Most current multi-modality learning frameworks, and even the broader landscape of machine learning systems, rely on a learning paradigm that differs substantially

from those observed in human cognition. When exposed to new knowledge, we instinctively form a concept and associate it with the external stimuli tied to that information. This newly formed concept is then integrated into our existing body of knowledge and stored as persistent memory in the mind. In contrast, most machine learning frameworks encode knowledge into large sets of model parameters that are difficult to interpret without specific model input. As a result, the activation of learned knowledge in such systems is often transient and dependent on specific inputs. This fundamental difference presents a significant challenge in designing systems that can explicitly form, retain, and reason over interpretable concepts in a manner analogous to human cognition.

The inclusion of a structured concept space and the use of concept-grounded inference may initially appear restrictive, particularly when compared with conventional models that rely on dense, task-specific representations optimized end to end for performance. However, we view this design as a deliberate and principled choice. By introducing a concept space that reflects structured, abstract knowledge similar to how humans form and retain concepts, the framework gains several benefits that are otherwise difficult to achieve. These include more efficient learning, natural generalization across modalities, and interpretability through explicit probing. The concept space acts as an inductive bias consistent with human cognition, enabling machine learning systems to operate in a more principled and cognitively grounded manner. We believe this work highlights a compelling direction for rethinking learning systems to more closely mirror human intelligence.

**Addressing Bias.** Hidden biases learned from datasets often hinder the trustworthiness of ML systems (Amodei et al., 2016; Lederer, 2023; Kaur et al., 2022; Knott et al., 2023). For example, NLP models often tend to associate the word "monarch" more with the word "male" than "female," reflected, for instance, in higher similarity scores between embeddings of "monarch" and "male." Our proposed framework facilitates effective probing into the model's learned knowledge and offers the capacity to rectify such biases.

| Concept 1 | Concept 2 | Concept Space | Ground Truth |
|---|---|---|---|
| Orange | Bus | 0.043 | 0.043 |
| Old | Building | 0.032 | 0.048 |
| Smiling | Person | 0.074 | 0.073 |
| White | Snow | 0.910 | 0.974 |
| Parked | Car | 0.228 | 0.244 |
| Cloudy | Sky | 0.173 | 0.192 |

**Table 6:** Sample Entailment Relation Queries of Concepts in Learned GQA Concept Space

Table 6 shows probing of a learned concept space fitted to the GQA dataset in action. Our framework enables easy querying of targeted concept pairs, which would be computationally expensive, if not infeasible, in traditional latent spaces. Further demonstrations of probing into the learned concept space can be found in Appendix A.3.

Revisiting the earlier example of the concept pair "monarch" and gender, the bias can be addressed directly in our framework by adjusting the ground-truth entailment probabilities. Specifically, ensuring equal entailment probabilities between "monarch–male" and "monarch–female" mitigates representational bias, a correction that can be easily applied through user-guided specification.

**Scalability of the Concept Space.** In our experiments, the concept space is constructed to reflect ground-truth entailment probabilities observed in training data. This approach can scale to larger and more diverse sets of concepts. Prior work (Vilnis et al., 2018; Li et al., 2018; Lai & Hockenmaier, 2017) has shown that similar embedding structures can learn entailment relations for large ontologies such as WordNet (WordNet). Although scaling introduces challenges in generating ground-truth probabilities, textual corpora offer a promising resource for extracting such relations, as demonstrated by He & Peng (2020). To assess scalability, we fitted a concept space to the full set of WordNet noun entries, totaling 10,765 concepts. The resulting space achieved a KL divergence of 0.1308 with respect to the ground truth, compared to 0.1172 for the GQA concept space.

**Call for Concept-Focused Datasets.** A major bottleneck in concept-centric learning is the lack of high-quality datasets with accurate concept annotations. In our experience, even after preprocessing, concept and attribute labels in datasets such as COCO and GQA contain substantial noise. This limits not only the performance of our framework but also that of other systems. Recent works (Pham et al., 2021; Saini et al., 2022; Bravo et al., 2023; Vedantam et al., 2020) have begun addressing this gap by improving the collection and annotation of attribute-focused datasets. We believe that further efforts to build datasets with richer and more reliable concept annotations will greatly support the development of interpretable and trustworthy AI systems.

**Integrating Concept Spaces with Vision-Language Models.** Compared to conventional vision-language foundation models, our cognitively inspired framework maintains an explicit and interpretable concept space for retaining and reasoning with knowledge, but its reasoning capabilities are currently limited to more controlled and simpler task domains. In contrast, modern VLMs, while operating largely as black boxes, offer strong flexibility and broad generalization across diverse downstream tasks through large-scale pretraining. A promising direction for future work is to explore hybrid architectures that combine these strengths. A concept space, which could take a form similar to that adopted here or extend beyond capturing only entailment relations, could be incorporated into a transformer-based VLM as a set of learned model parameters. Attention mechanisms can then be used to associate model inputs with the knowledge embedded in the concept space. These concept parameters could be kept frozen, allowing the model to use them as a stable semantic memory, or updated during downstream training, enabling the model to refine or even discover new concepts through interaction with task data.

**Scope and Limitations.** In the current framework, the concept space supports only two types of concepts: attributes and categories. Future work should explore methods to expand this space to a broader range of concepts, such as abstract or relational concepts for more complex reasoning, and action-oriented concepts to accommodate modalities beyond vision and language, such as robot planning.

## 6 Conclusion

Human cognition exhibits a remarkable ability to form a coherent and structured understanding of the world, and to apply this knowledge efficiently across diverse tasks and modalities. Inspired by this capability, we propose a concept-centric multi-modality learning framework centered around a modality-agnostic concept space that captures universally applicable knowledge.

The primary technical contribution lies in the design of a modular framework that integrates a shared concept space with a flexible set of modality-specific projection models. This design enables knowledge reuse across modalities and task domains, supporting learning that is interpretable, generalizable, and modular. Unlike traditional end-to-end learning systems that encode knowledge implicitly within dense parameter spaces, the proposed framework embeds knowledge explicitly into a structured concept embedding space, enabling interpretability through efficient probing of concept entailment probabilities.

Empirically, the experiments demonstrate that the proposed framework supports more efficient learning. In the vision modality, the projection model converges significantly faster than a baseline model built on a traditional architecture. This gain in efficiency is attributed to the fact that the concept space already encodes structured, abstract knowledge that the projection model can adapt to. Additionally, we evaluated the framework on two downstream tasks, Image-Text Matching and Visual Question Answering, and demonstrated that our method achieves performance comparable to state-of-the-art methods even without task-specific fine-tuning. While our goal is not to surpass existing benchmarks in raw performance, these results support the viability of a cognitively inspired learning paradigm. Rather than optimizing solely for accuracy, our framework emphasizes learning efficiency, interpretability, and structural alignment with human cognition. These qualities are increasingly important as machine learning systems are deployed in more complex and dynamic environments.

More broadly, the work motivates a rethinking of how machine learning systems acquire and represent knowledge. By introducing a concept space as an inductive bias, this framework opens a promising research

direction toward building systems that more naturally align with human reasoning. Such systems may offer greater transparency, flexibility, and the ability to generalize knowledge across tasks and modalities.

Looking forward, several directions can further extend the proposed framework. First, scaling the concept space to support larger vocabularies and richer relational structures beyond entailment, including compositional and causal relations, would expand its expressive power. Second, applying the framework to new task domains such as concept-grounded Text-to-Image generation presents a natural extension. Beyond these extensions, the results point toward meta-learning approaches that enable learning systems to discover, update, and organize concepts directly from multimodal inputs, including vision and language. Rather than treating concept vocabularies as fixed and externally provided, future systems should be able to infer and refine conceptual structures from raw data as models interact with new modalities and tasks. Advancing adaptive concept learning mechanisms for concept discovery, concept organization, and concept application is an important step toward more autonomous, scalable, and cognitively grounded machine learning systems. Finally, explicit concept-level representations may serve as a semantic interface for semantic communication, in which agents exchange meaning at the level of concepts and relations rather than raw signals or low-level features.

## Acknowledgements

This work was supported in part by the National Science Foundation under Grant No. 2133403, and by computing credits provided by Amazon Web Services and Microsoft Azure. We thank the reviewers and the action editor for their thoughtful feedback and constructive suggestions.

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

# A Concept Space Details

## A.1 Preliminary

A smoothing function for the concept space is defined as:

$$m^i_{\text{soft}}(\omega) = \frac{\text{softplus}(\omega^i)}{\text{softplus}(G^i_{max} - G^i_{min})} \tag{4}$$

where the denominator is a normalization term with $G_{max}, G_{min}$ being the global maximum and minimum values at $i$ dimension. In short, this smoothing function is introduced so a valid joint probability can be calculated even if two concepts/boxes are disjoint and we refer readers to Li et al. (2018) for its complete proof.

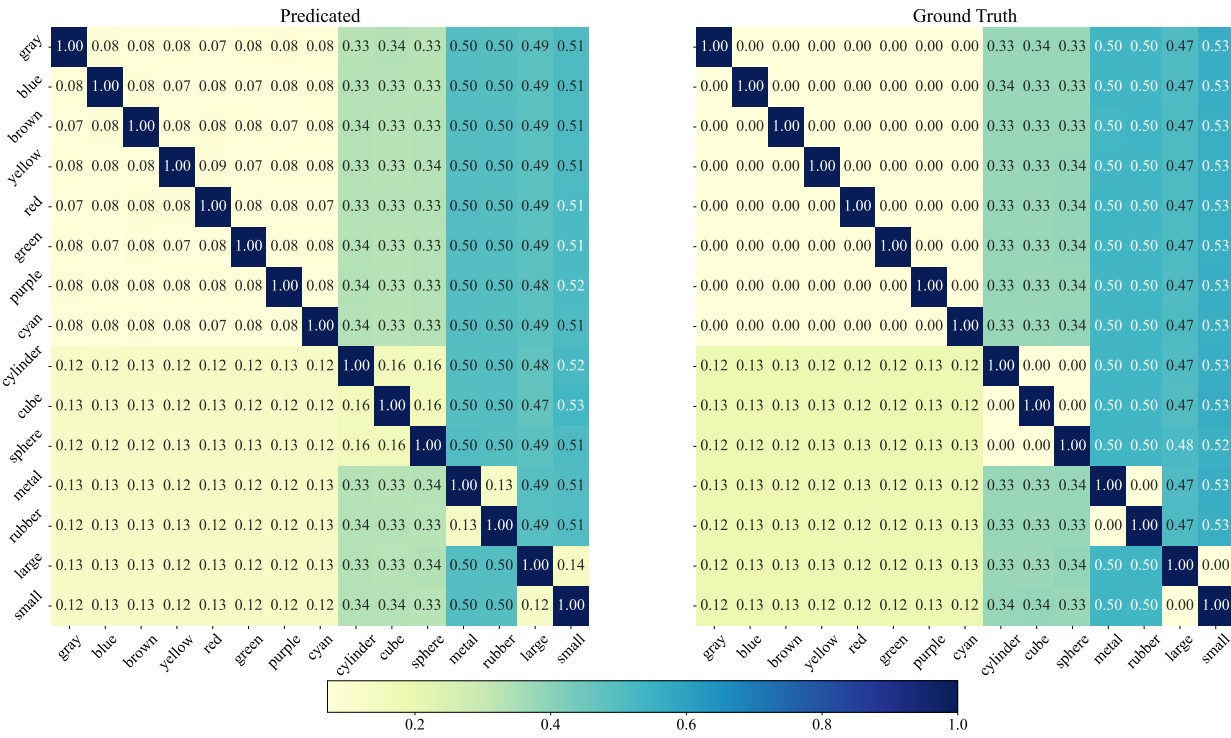

**Figure 5:** A comparison between the learned concept space's understanding of the CLEVR world and the ground truth relations illustrated via entailment probabilities of concept pairs. Such comparison allows simple probing into the knowledge learned by this abstract concept space. A SoftMax function is applied on entailment probabilities of same-attribute concepts conditioned on a single concept $y$ so $\sum_{y' \in \text{attr}_i} P(y'|y) = 1$ is satisfied.

## A.2 Concept Space Training Objective

We define a KL-divergence measure between a predicted conditional probability distribution $q(y_1|y_2)$ and a target $p(y_1|y_2)$ as:

$$D_{\mathbf{KL}}(P(y_1|y_2)||Q(y_1|y_2)) = \mathbb{E}_{(y_1,y_2)\sim P}\left[\log \frac{P(y_1|y_2)}{Q(y_1|y_2)}\right] \tag{5}$$

Let $\binom{\boldsymbol{y}}{2}$ denote a set of all concept pairs created from 2-combination from $\boldsymbol{y}$ The objective for training the concept space is formally described as the following:

$$\mathcal{L}_{\text{concept}}(\mathcal{C};\mathcal{D}_*) = \frac{1}{|\mathcal{D}_*|}\sum_{(x,\boldsymbol{y})\in\mathcal{D}_*}$$
$$\frac{1}{2\cdot\left|\binom{\boldsymbol{y}}{2}\right|}\sum_{(y_1,y_2)\in\binom{\boldsymbol{y}}{2}} D_{\mathbf{KL}}(P(y_1|y_2)||Q(y_1|y_2)) + D_{\mathbf{KL}}(1-P(y_1|y_2)||1-Q(y_1|y_2)) \tag{6}$$

## A.3 Probing into Concept Space

Figure 5 shows an example of probing into learned knowledge of the concept space exposed to CLEVR. Benefited from such efficient probing mechanism, this concept space offers more interpretability compared to traditional latent spaces or model parameters of previous learning frameworks.

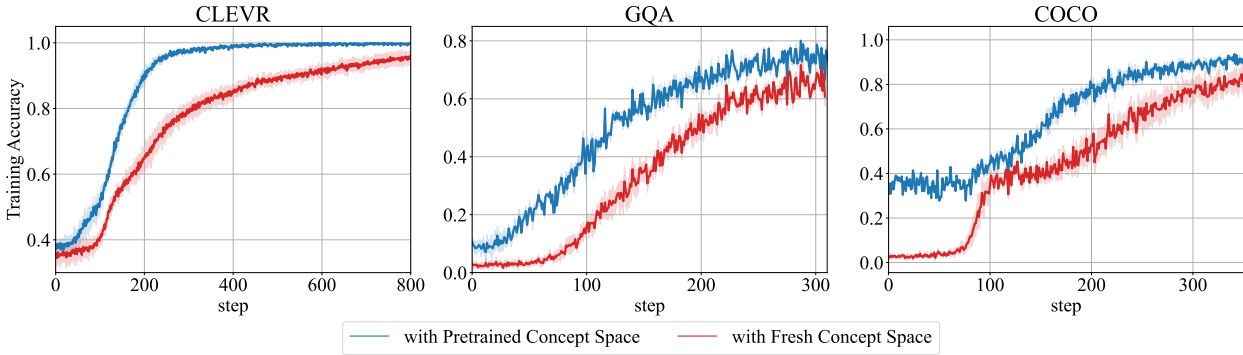

**Figure 6:** Ablation study on the pretrained concept space. We cut our projection models' access to the pretrained concept space and the learning of this concept space is combined into training processes of the projection models. Shaded area in plots represents 2-sigma error over five trails of experiments. Their classification accuracy is used to compare the ablated version and the original framework.

| Dataset | Dim ($K$) | Accuracy | mAP |
|---------|-----------|----------|-----|
| | 24 | 0.99898 | – |
| CLEVR | 50 | **0.99900** | – |
| | 96 | 0.92914 | – |
| | 24 | 0.95899 | 0.57212 |
| COCO | 50 | **0.95653** | **0.58961** |
| | 96 | 0.56060 | 0.56024 |
| | 24 | 0.84752 | 0.21271 |
| GQA | 50 | **0.84419** | **0.35202** |
| | 96 | 0.61458 | 0.32331 |

**Table 7:** Ablation on concept space dimensionality ($K$). CLEVR reports accuracy only; COCO and GQA report accuracy and mAP. Bolded values indicate the selected dimensionality ($K = 50$).

## A.4 Ablation on Concept Space

We discover that using a pretrained concept space with learned abstract knowledge helps modality-specific projection models converge faster compared to the ones without the access. Specifically, we cut our framework's access to the pretrained concept space $\mathcal{C}$. Instead, the framework is only provided with a freshly initialized concept space $\mathcal{C}'$ and the loss function during pretraining of the vision-modality projection model is changed to $\mathcal{L}'_{\text{vision}} = \mathcal{L}_{\text{vision}} + \mathcal{L}_{\mathcal{C}}$. Fig. 6 shows that the original framework's projection models can converge faster than the ablated version. Based on this evidence, we conclude that the abstract knowledge shared by the pretrained concept space streamlines the learning process of modality-specific projection models.

## A.5 Ablation on Concept Space Dimensionality

To assess how the dimensionality of the concept space affects downstream performance, we conduct an ablation study using $K \in \{24, 50, 96\}$. For each value of $K$, we follow the same pretraining protocol: we first train the concept space to convergence and then train the ViT-based projection model to adapt to that space. All runs use identical hyperparameters, optimization settings, and training steps to ensure a fair comparison. Results are reported in Table 7.

Empirically, $K = 50$ achieves the strongest and most consistent performance across datasets, showing the highest mAP on both COCO and GQA while matching the near-saturated accuracy on CLEVR. These

| Dataset | # Attributes | # Categories |
|---|---|---|
| GQA | 35 | 33 |
| CLEVR | 12 | 3 |
| COCO | 35 | 29 |
| Unified (All) | 70 | 60 |

**Table 8:** Summary of concept subsets used in each dataset.

observations provide practical support for choosing $K = 50$ as the concept-space dimensionality used in all main experiments.

We also observe that increasing the dimensionality to $K = 96$ leads to noticeable performance degradation, particularly on COCO and GQA. A plausible explanation is that larger box dimensions introduce a more complex geometric structure, which makes the concept space harder to learn effectively and can lead to diminishing returns or overfitting. Conversely, very small dimensions, such as $K = 24$, limit expressivity and may restrict performance on semantically richer datasets. The intermediate dimensionality of $K = 50$ therefore offers a favorable balance between expressiveness and learnability.

## B   Cross Modality Joint Training

To allow probabilistic analysis for cross-modality tasks, an *optional* joint training stage can be used to encourage different projection models to produce projections that overlap with each other's for the same object. This joint training stage is optional because each individual projection model is already adapted to the shared knowledge space, and it is computationally lightweight, as the modality-specific projection models have already been trained and aligned with the unified concept space. It requires very modest resources, with convergence occurring within a few hundred training steps, as demonstrated in Sec. 4.2. Subsequently, this design with demonstrated efficiency allows the effortless incorporation of new projection models into our proposed framework, mirroring humans' ability to learn and link knowledge across modalities in a fast and efficient manner. Specifically, consider a system with two modalities, A and B, as an example. The training dataset would be denoted as $\mathcal{D}_{A \cup B} = \{(x_i^A, x_i^B, \boldsymbol{y}_i)\}_{i=1}^{N}$, and the training objective for this joint training stage is defined as:

$$
\mathcal{L}_{\text{joint}}(\theta_A, \theta_B; \mathcal{D}_{A \cup B}) = \frac{1}{2|\mathcal{D}_{A \cup B}|} \sum_{(x_A, x_B, \boldsymbol{y}) \in \mathcal{D}_{A \cup B}}
$$
$$
P(f_A(x_A; \theta_A) \mid f_B(x_B; \theta_B)) + P(f_B(x_B; \theta_B) \mid f_A(x_A; \theta_A))
$$
(7)

The overall training objective becomes a combination of modality-specific projection losses and this joint training loss.

## C   Evaluation Datasets and Preprocessing

### C.1   Dataset Details

We base our evaluations on three datasets and a world dataset aggregating all concepts and representations together:

**CLEVR** dataset comprises synthesized images paired with intricate questions testing a system's visual reasoning capabilities. We choose CLEVR for evaluation because it provides a highly controlled mini-world, where concepts are easily drawn from visual objects, and relationships between concepts are clearly defined. Each CLEVR image displays a scene with a random number of objects, each described by `color`, `shape`, `material`, and `size`, which produces 15 unique values such as `blue` and `cube`, forming attribute concepts related to specific objects.

| Type | CLEVR | COCO | GQA |
|------|-------|------|-----|
| **Attributes** | blue, brown, cyan, gray, green, large, metal, purple, red, rubber, small, yellow | adult, appetizing, athletic, busy, casual, cloth, cooked, enjoying, family-friendly, female, fluffy, fresh, functional, furry, hairy, healthy, holding, horizontal, laying, male, metal/metallic, moving, parked, participating, public, sitting, socializing, soft, sporty, standing, tame, tasty/delicious, useful, vertical, watching/looking | black, blue, brick, brown, clear, cloudy, concrete, dark, glass, gray, green, happy, large, long, metal, old, open, orange, parked, pink, red, round, short, silver, sitting, small, smiling, standing, striped, tall, tan, white, wood, yellow, young |
| **Categories** | cube, cylinder, sphere | airplane, apple, banana, bear, bicycle, bird, boat, broccoli, bus, cake, car, carrot, cat, cow, dog, donut, elephant, giraffe, horse, hot dog, motorcycle, orange, person, pizza, sandwich, sheep, train, truck, zebra | bed, boy, building, bus, car, chair, fence, field, floor, giraffe, girl, grass, ground, hair, head, jacket, man, person, plate, road, shirt, sidewalk, sky, snow, street, table, train, tree, trees, wall, water, window, woman |

**Table 9:** Exact attribute and category concepts present in each dataset.

| Concept | Type | Datasets |
|---------|------|----------|
| blue | attribute | CLEVR, GQA |
| brown | attribute | CLEVR, GQA |
| gray | attribute | CLEVR, GQA |
| green | attribute | CLEVR, GQA |
| large | attribute | CLEVR, GQA |
| metal | attribute | CLEVR, GQA |
| parked | attribute | COCO, GQA |
| red | attribute | CLEVR, GQA |
| sitting | attribute | COCO, GQA |
| small | attribute | CLEVR, GQA |
| standing | attribute | COCO, GQA |
| yellow | attribute | CLEVR, GQA |
| bus | category | COCO, GQA |
| car | category | COCO, GQA |
| giraffe | category | COCO, GQA |
| person | category | COCO, GQA |
| train | category | COCO, GQA |

**Table 10:** Overlapping concepts appearing in multiple datasets. These concepts are merged in the Unified dataset.

**COCO** dataset exposes our framework to a knowledge world resembling the real world better than computer-generated images from CLEVR. We use attribute annotations proposed by Patterson & Hays to establish attribute concepts such as `soft`, `cooked`, and `parked` (see Fig. 1 in (2016)). The original COCO object categories are used as category concepts. We focus our evaluation on the top 35 frequent attributes and their associated categories to gain meaningful insights, resulting in 64 concepts (see Table 9).

**GQA** dataset is similar to COCO, providing a controlled sandbox mimicking real-world settings. We use the original attribute and category labels in GQA as concepts and filter out rare attributes and classes, resulting in the same total number of concepts as COCO. Example attribute and category concepts include `happy`, `old`, `gray`, and `boy`. Exact concepts can be found in Table 9.

**Unified** dataset aggregates all attribute and category concepts appearing in CLEVR, COCO, and GQA into a single world dataset, resulting in 130 unique concepts (70 attributes and 60 categories; see Table 8). Overlapping concepts across datasets (listed in Table 10) are merged into unified concepts. This dataset allows us to evaluate the scalability of the proposed framework and its ability to generalize abstract concepts across different visual and linguistic domains. To avoid the Unified dataset being dominated by CLEVR, which contains significantly more samples than COCO and GQA, we balance the dataset sizes during construction. In the raw training splits, CLEVR contains 455,632 samples, compared to 78,898 in GQA and 91,667 in COCO. We therefore cap the CLEVR training split to match the size of the largest non-CLEVR dataset (COCO), resulting in 91,667 CLEVR samples after filtering. This ensures that the Unified dataset remains balanced and diverse across data sources.

### C.2 Dataset Preprocessing

Since each image in these datasets contains multiple objects, a preprocessing step is essential to isolate single objects. This isolation allows focused learning on targeted objects, reducing ambiguity. This process mirrors human learning, where attention naturally centers on a novel object while ignoring the surrounding environment Gärdenfors (2014).

Both COCO and GQA datasets already include object segmentation data. For the CLEVR dataset, we employ a MASK R-CNN model (He et al., 2017), denoted as $f_{\text{detection}}$, trained on a small amount of annotated data as an object detection model to generate segmentation. Visual object inputs are created by cropping original images to include only the objects of interest, as illustrated in Fig. 7.

In addition to object isolation, we generate a descriptive sentence for each object, introducing natural language as a new modality in the dataset. Each sentence of an object has the structure "*There is a*" followed by a sequence of values indicated by its attribute concepts in random orders to ensure diversity. Category concept values are added last to the sequence, except for CLEVR, where values from the `shape` attribute family are placed last for natural-sounding sentences.

## D Model Details

### D.1 Backbone Architectures

#### D.1.1 Vision Modality

**ViT.** A Vision Transformer Dosovitskiy et al. (2020) pretrained on ImageNet-21k (`vit-base-patch16-224`) is used as a backbone. The pooled embedding at the `[CLS]` token is passed to either the projection head (our framework) or the MLP head (baseline).

**ResNets.** A ResNet-50 He et al. (2015) pretrained on ImageNet-21k is used as a backbone. Its final fully connected layer is replaced by either our projection head or the baseline MLP head.

#### D.1.2 Natural Language Modality

**BERT.** A pretrained BERT-Base encoder Devlin et al. (2018) is used as the backbone for the natural-language modality. Analogous to the ViT setup, the pooled representation at the `[CLS]` position is used as input to both the projection head and the MLP head.

### D.2 Baseline MLP Heads

To provide consistent and comparable baselines for our projection models, we attach a simple three-layer MLP head to each backbone, whereas our projection heads consist of only a single linear layer. Unless otherwise

specified, the two intermediate layers contain 128 units each. For ResNet-based models, however, the baseline MLP head uses 512 and 256 units in the intermediate layers to accommodate the higher dimensionality of ResNet's feature representation.

**MLP Head Output Dimension.** The output dimension of each baseline MLP is set to the total number of concepts present in the dataset:

$$\dim(\text{out}) = |\mathcal{Y}_{\text{category}}| + |\mathcal{Y}_{\text{attribute}}|.$$

Final predictions are obtained by applying an `argmax` over the category neurons and a threshold over the attribute neurons.

### D.3 Projection Heads

As shown in Algo. 1, our projection head is implemented using two parallel linear layers. Conceptually, the projection head is a single linear mapping from the backbone feature vector into the knowledge space $\mathcal{K}$. For computational convenience, however, we split the backbone feature vector into two equal-sized chunks and apply an independent linear layer to each chunk. This is equivalent to applying one linear layer to the full feature vector, but allows us to separately obtain $(\omega_{\min}, \omega_{\Delta}) \in \mathcal{K}$ in our framework.

### D.4 Training Details

Vision modality projection models are trained for 10 epochs with a batch size of 256 with an exception of CLEVR whose models are only trained for 1 epoch. An AdamW optimizer with a learning rate of $10^{-4}$ is used. Learning rate schedulers are used to achieve warm-up for first epoch and then a process of $10^{-1}$ linear decrease over the remaining epochs.

Natural-language modality projection models are trained for 1 epoch using the same setup and hyper-parameters as used by the vision ones.

Thresholds for attribute identification are selected based on performances from training splits. Thresholds producing the best f1 score on training sets are used in tests.

## E Image-Text Matching Experiment Details

### E.1 BLIP

We follow the training method as stated in Li et al. (2022) and fine-tune the pretrained BLIP model directly on the Image-Text Matching task (swapping-sentence split) using both the image-text contrastive loss and a task-specific image-text matching loss produced by the image-text matching classification head in BLIP. We use a greater batch size of 512 as the calculation of image-text contrastive loss requires a large number of samples.

### E.2 CLIP

We follow the training method as stated in Radford et al. (2021) and adapt the pretrained CLIP model to the general three datasets using the symmetric loss that favors larger similarity scores between positive image-text pairs and smaller scores for negative ones. We use a batch size of 512 as in BLIP during pretraining. Similar to our framework, CLIP model is not directly trained on the Image-Text Matching task.

### E.3 ViLT

Similar to BLIP, we follow the training method as stated in Kim et al. (2021b) and fine-tune the pretrained ViLT model directly on Image-Text Matching task (swapping-sentence split) using a binary cross-entropy loss on the matching classification head.

### E.4 FLAVA

We use the same procedures as used in ViLT to fine-tune a pretrained FLAVA model on the data domains appeared.

## F  Additional Figures

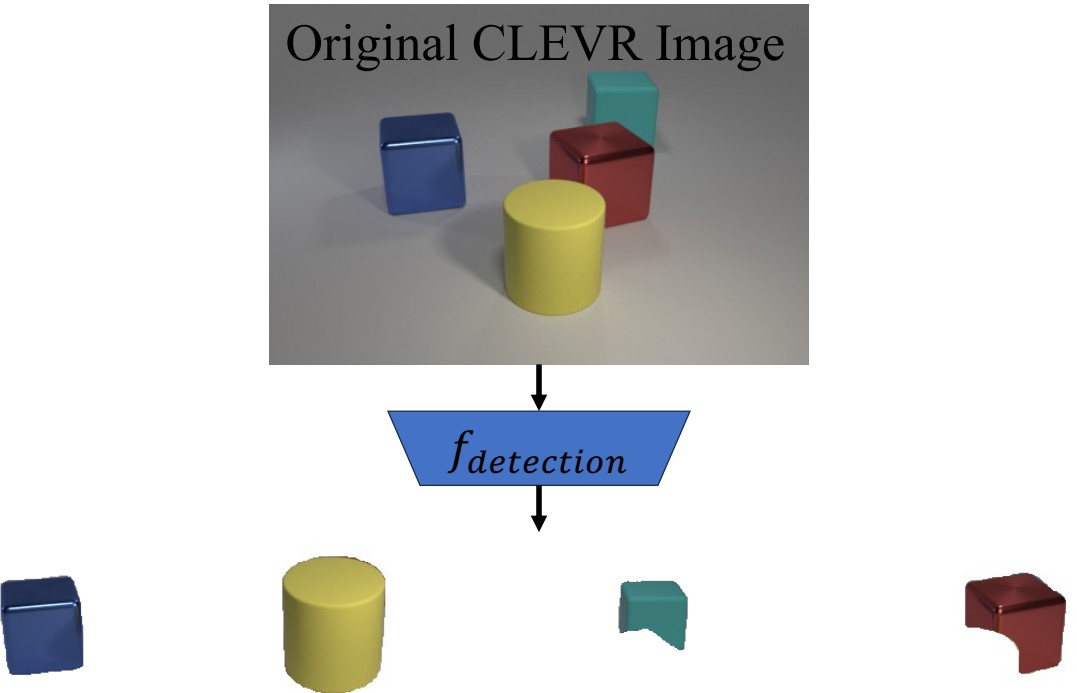

**Figure 7:** The segmentation masks generated by $f_{\text{detection}}$ are applied to the original CLEVR images to isolate each object from its surroundings environment. This preprocessing step enables our proposed framework to replicate the way we, as humans, naturally focus our attention on novel objects during the learning process.

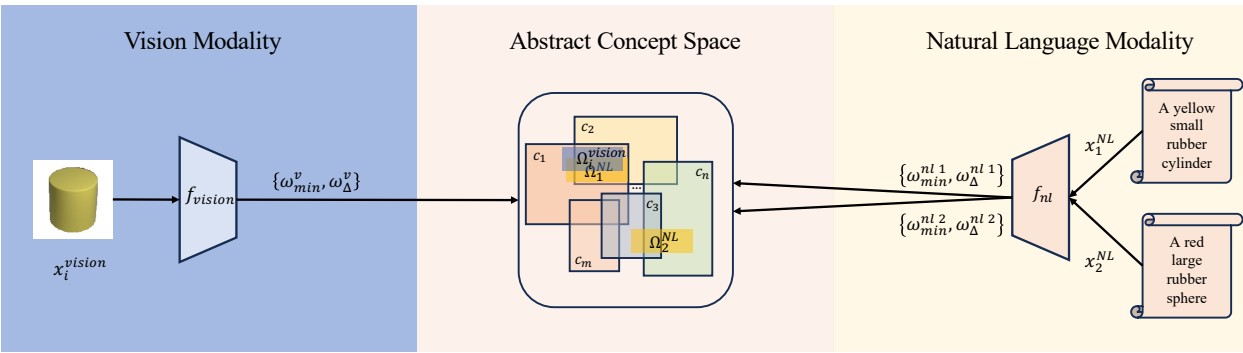

**Figure 8:** Application of the proposed framework on the Image-text matching task. An image $x_i^{\text{vision}}$ of a yellow, small rubber cylinder and two description sentences $x_1^{\text{NL}}, x_2^{\text{NL}}$ are processed by their modality-specific models $f_{\text{vision}}$ and $f_{\text{NL}}$ which project modality-specific inputs onto a learned abstract concept space $\mathcal{C}$. We use the cross-entailment probability between projections of an image and a sentence to determine if they form a positive pair. While creating representations of images and sentences in a shared latent space is a common approach for the image-text matching task, our shared representation space is a knowledge-embedded concept space offering interpretability, which is in drastic contrast to the commonly used latent space with black-box structure.

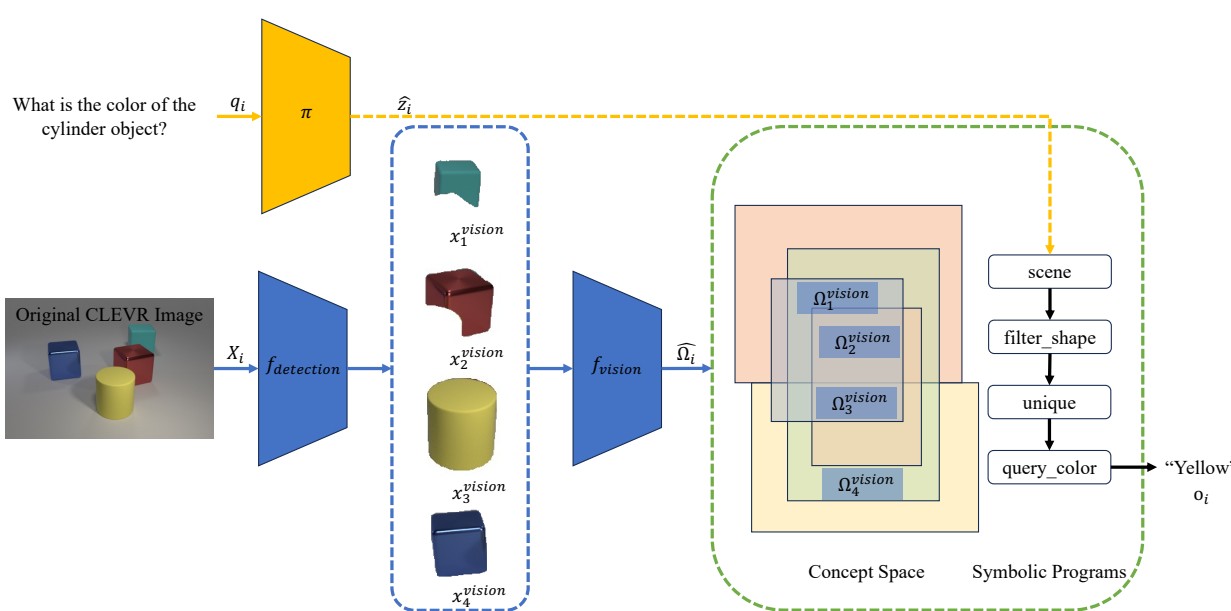

**Figure 9:** Application of the proposed framework to Visual Question Answering task. We reuse the object detection model $f_{detection}$ from the pretraining stage, which extracts a set of single objects $\boldsymbol{x}_i$ from an original CLEVR image $X_i$. The vision-modality projection model $f_{\text{vision}}$ then projects $\boldsymbol{x}_i$ onto the $\mathcal{K}$. A program generator $\pi$ is used to predict a sequence of symbolic programs $\hat{z}_i$ based on an input question $q_i$ in natural language format. Programs in $\hat{z}_i$ operate on the concept space and produce an answer $o_i$ to $q_i$.

