# OpenReview forum: "A Concept-Centric Approach to Multi-Modality Learning"
_TMLR — Accepted by TMLR_

### Review · Reviewer_5zbu · 2025-10-23

**Summary Of Contributions:**

Strengths:
1. The submission proposing a new direction to align knowledge from different modalities by aligning them in a common space with concepts. The motivation of this concept-centered approach itself is promising and reasonable.

Weaknesses:
1. The work straightforwardly extends the previous box embedding solution to the multimodal case. Unfortunately, some key challenges about multimodality are not well-explained. For example, 1) How to evaluate whether the learned concepts are modality-agnostic; 2) How to design general decoders to utilize the modality-agnostic concepts.
2. The experimental results and evaluations are not comprehensive. Some baselines are relatively out-of-date. For example, in the VQA task, the latest baseline in published in 2021. Meanwhile, the performance of the proposed method doesn't show better/stronger gains, which greatly limits the illustration of the proposed method.
3. The presentation can be further improved, and lots of details are missing, such as the specific design choices for various potential downstream applications.

**Audience:**

Yes

**Audience Explanation:**

The idea of concept-centric multimodal learning itself is an interesting direction. Thus, it may still be of interest for some individuals.

**Broader Impact Concerns:**

N.A.

**Claims And Evidence:**

No

**Claims Explanation:**

The submission claims a very general and intuitive framework for multimodal learning. However, the experimental results on two simple tasks and comparisons with out-of-date baselines don't demonstrate the claimed strengths of the general concept-centric framework.

**Requested Changes:**

1. More comprehensive evaluation to demonstrate the effectiveness and generalization of the proposed framework.
2. Polish the presentation by adding more details and descriptions about the methodology.

---

> ### Author Response · Authors · 2025-11-19
> **Response to Reviewer 5zbu**
>
> We sincerely appreciate reviewer 5zbu for their constructive feedback on our work. We are also motivated that the reviewer finds the motivation of our concept-centered framework promising and reasonable. Please find our changes and responses below.
>
> ## Changes made to the paper
>
> 1. More comprehensive evaluation
>
>     We appreciate the reviewer’s suggestion on adding more experiments for a more complete evaluation and have made the following additions to our revised paper.
>     - New dataset for more comprehensive evaluation:
>
>         We have included a new unified dataset that combines all concepts and representations from CLEVR, GQA, and COCO. This unified dataset introduces a more challenging and informative setting, where overlapping concepts appear across different data domains and the framework must link representations from these different domains to the *same* underlying concepts. It also contains a substantially larger set of 130 concepts. A detailed description of this dataset is provided in Section C1 and Table 10 on page 23 of the revised paper.
>     - Added mAP metric for further demonstration of  effectiveness:
>
>         In addition to the new dataset, we have updated our Table 1 and expanded Figure 2 on page 8 of our revised paper to report the mAP for our vision-modality projection's performance on attribute concepts.
>     - New experiment on evaluating cross-modality alignment:
>
>         We have added a new experiment in Section 4.2 demonstrating cross-modality alignment for our independently trained vision and text projection models. The results in Table 2 on page 9 show strong alignment of the projection models in our framework without any joint fine-tuning, which stands in contrast to mainstream multimodal models that require explicit joint training to achieve alignment.
>
>         Then, only for the purpose of illustrating cross-modality alignment efficiency, we apply an optional joint-training objective to finetune the projection models in our framework and compare it with benchmark multimodal models in Figure 3. The results show that our projection models require significantly less GPU time during the finetuning stage.
>
>     - New experiment on evaluating cross-modality generalization via acquisition of new modalities:
>
>         We have added a new experiment in Section 4.3 to evaluate the cross-modality generalization of our framework through the acquisition of three additional modalities. In this experiment, the natural language descriptions originally used for the English modality are translated into Chinese, French, and Spanish, creating three new modalities on top of the existing vision and English modalities.
>
>         All five projection models are trained independently with no joint training. We use a cross-modality representation-matching task as a proxy for assessing alignment in the shared knowledge space. The results in Table 3 show that the newly added modalities exhibit strong alignment with the existing ones as well as with each other. To provide further insight into their behavior, Figure 4 reports the entailment probabilities of their projections for positive and negative cross-modality representation pairs.
>
>         This experiment illustrates the scalability of the framework: incorporating a new modality requires only training a corresponding projection model to adapt to the universal knowledge space. No modification or fine-tuning of existing projection models is needed, which stands in contrast with many mainstream multimodal approaches.

---

> ### Author Response · Authors · 2025-11-19
> **Response to Reviewer 5zbu (cont’d)**
>
> ## Changes made to the paper (cont'd)
>
> 2. Polish the presentation by adding more methodological details
>
>     We agree with the reviewer that the presentation can be further polished, and we have made the following improvements in our revised paper.
>     - We added explicit description of the probabilistic box embedding space and the geometric intuition behind entailment probabilities in Section A1 on page 19.
>     - We added specific details regarding CLEVR, GQA, and COCO in Tables 8, 9, and 10 in the appendix.
>     - We added explanation for negative sampling on page 5 to clarify why it is necessary when training the concept space.
>     - We added clarification on how spatial information is obtained for VQA in the Object Detection and Projection section on page 12.
>     - We added description of how the Program Generator is trained on page 11 and clarified that we followed the same pretraining setup instead of using a pretrained model directly.
>     - We moved the optional joint training section to the appendix and clearly indicated that joint training is used only in Sections 4.2 and 4.3 for illustration of cross-modality efficiency. All other results, including VQA, are produced without fine-tuning existing models.
>     - We added citation to COCO Attributes in the dataset section on page 6 and added reference to Figure 1 from that paper in the dataset details section on page 23.
>     - We added a discussion segment in Section 5 on page 13 about potential ways to integrate the proposed framework with more widely adopted vision-language foundation architectures.
>
> ## Clarifications
>
> Now, we would like to offer some clarifications regarding the reivwer's questions.
>
> - Addressing modality-agnostic concepts and decoder design
>
>     We appreciate the reviewer’s thoughtful questions regarding (1) how to assess whether the learned concepts are modality-agnostic and (2) how general decoders might be constructed to make use of these concepts. We clarify both points below.
>     - Evaluating whether the concepts are modality-agnostic.
>
>         In our framework, concepts are not learned from any specific modality’s representations. Instead, they correspond to high-level semantic entities, such as attributes and categories, whose entailment structure is learned directly from the dataset’s concept annotations. Because the concept space is trained without using any modality-specific embeddings, the resulting concept boxes encode semantic and entailment relations that remain abstract and modality independent.
>
>         To help clarify this point, we also highlight the empirical behavior of the framework. As shown in our revised experiments, the projection models trained on different modalities naturally align with each other in the shared concept space even though they are trained independently. This consistent alignment across vision, English, and three additional languages reflects the fact that the learned concepts are not tied to any particular modality and instead capture modality-agnostic semantic structure.
>
>     - Designing general decoders that use the learned concepts.
>
>         We agree that designing universal decoders capable of mapping modality-agnostic concepts back into modality-specific outputs raises an important research direction. It connects naturally to broader topics such as grounding, controllable concept-guided generation, and concept-to-modality reconstruction. While developing such decoders is beyond the scope of the current paper, we appreciate the reviewer bringing attention to this direction. Our revised discussion on pages 13–14 already includes several use cases and examples of concept-grounded generation, and we have expanded that section to more clearly articulate potential strategies and future opportunities in this space. We see this as a promising direction and are actively exploring it.

---

> > ### Author Response · Authors · 2025-11-19
> > **Response to Reviewer 5zbu (cont’d - 2)**
> >
> > ## Clarifications (cont’d)
> >
> > - Addressing experiment resutls and comparison
> >
> >     We appreciate the reviewer’s concern regarding baseline recency and performance comparisons. We would like to clarify that the goal of this work is not to pursue task-specific optimization or to outperform heavily engineered multimodal models. As outlined in both the abstract and the discussion section on page 12 (“A Cognition-Inspired Learning Paradigm”), the goal of our work is to introduce a cognitively inspired learning scheme that emphasizes efficiency, interpretability, modularity, and concept-centric reasoning. The aim of the experiments is therefore not to maximize task accuracy through extensive model-specific fine-tuning, but to illustrate how these properties emerge within our proposed framework.
> >
> >     In the VQA experiments, our approach does not apply any additional fine-tuning or task-specific adaptation to the concept space. The concept space learned in our framework remains unchanged when it is applied to downstream tasks. This design is fundamentally different from existing VQA models that rely on extensive joint optimization of for example program generator and reasoning model. Because our framework aims to demonstrate that reasoning can be performed in a modality-agnostic abstract space, rather than within large opaque representation embedding spaces or via LLMs, we expect task results to be comparable but not necessarily superior to highly specialized models.
> >
> > ---
> >
> > We sincerely thank reviewer 5zbu again for their thoughtful comments and for highlighting the potential impact of concept-centric multimodal learning. We are very happy to make any further adjustments that could improve the quality or clarity of the paper.

---

### Review · Reviewer_KSiN · 2025-11-06

**Summary Of Contributions:**

This paper proposes an alternative approach to multimodal representation learning that uses structured concept spaces rather than the dense vector representations employed in methods like CLIP. The framework consists of two stages: (1) learning a modality-agnostic concept space using probabilistic box embeddings that encode relationships between concepts via entailment probabilities, and (2) training modality-specific projection models that map inputs from different modalities into this shared concept space.

The core idea—using structured, interpretable concept spaces as an intermediate representation for multimodal learning—is interesting and well-motivated. However, the experimental validation does not convincingly support the broad claims made about efficient and modular multimodal learning and adaptation. The experiments are conducted in small-scale, controlled settings (CLEVR, simplified subsets of COCO and GQA with only 64 concepts) where the multimodal aspect is particularly weak: the natural language modality consists of synthetically generated, structured descriptions that are essentially concatenated lists of attributes, making it difficult to assess true cross-modal alignment capabilities.

## Strengths

- **Strong motivation and positioning:** The paper provides a compelling argument for concept-centric learning inspired by human cognition, with a thorough literature review connecting to relevant work in multimodal learning and concept learning.
- **Interesting technical approach:** The synthesis of structured embedding spaces (box embeddings), concept learning, and multimodal projection is novel and the use of supplementary concept annotations from datasets like COCO attributes and GQA scene graphs is well-executed.
- **Interpretability:** The framework demonstrates clear benefits for interpretable reasoning, particularly in the VQA experiments where symbolic programs operate over learned concept representations, and the ability to probe entailment probabilities between concepts is genuinely useful.
- **Thoughtful discussion:** The paper includes good consideration of future work, limitations, and broader implications including bias mitigation.

## Weaknesses

1. **[W-1] Claims Not Convincingly Supported by Experiments**
    - Multimodal claims are overstated: The paper claims to enable "efficient and modular multimodal learning" that bridges "the efficiency gap between traditional machine learning methods, which require extensive data, and human learning." However the text modality in all experiments consists of synthetically generated, highly structured descriptions (e.g. "There is a yellow small rubber cylinder") on small (<=64) concept spaces. While well motivated, it is hard to support claims of this breadth for multimodal learning.
    - Ablation studies do not sufficiently support core claims: The ablation study in Section 5 receives significant emphasis but does not effectively support the paper's central claims about efficient multimodal learning.

2. **[W-2] Presentation and Missing Details**
    - Important details are missing or difficult to find: Core technical and experimental details in Sections 3, 4 are either absent (impacting reproducibility) or inconsistently distributed between main text and appendices

3. **[W-3] Limited Scope and Missing Contextualization**
    - Relationship to current VLM paradigm not directly addressed: The paper does not discuss how this approach relates to, could complement, predominant vision-language model paradigm. For instance:
        - What would it take to handle more general scene understanding with complex natural language queries (as in full GQA) rather than controlled symbolic reasoning (as in CLEVR)?
        - Could this concept-space approach be integrated with or benefit from pretrained vision-text-language models?

**Overall**

The work would benefit from either (1) repositioning the claims to focus on the interpretability and structured reasoning benefits that are well-demonstrated, or (2) conducting additional experiments to make the claims more convincing (described below).

**Audience:**

Yes

**Audience Explanation:**

The paper addresses several questions of significant interest to the multi-modality learning and interpretable AI communities:
1. **Cognitively-inspired learning paradigm:** The concept-centric approach that mirrors human cognitive processes (forming abstract concepts and connecting them across modalities) represents a fundamentally different direction from current end-to-end learning systems. This perspective alone would interest researchers exploring alternatives to standard deep learning paradigms.
2. **Interpretability and probing:** The explicit concept space with queryable entailment probabilities (Table 4, Figure 4) offers a practical mechanism for model introspection that is notably absent in conventional latent spaces. This directly addresses growing concerns about ML system transparency and trustworthiness.
3. **Compositional and controllable representations:** The framework's treatment of concepts as modular, reusable entities aligns with important open questions in compositional generalization and systematic generalization. Moreover, the explicit concept space could enable better control and steering of generative models—a natural extension the authors note for concept-grounded text-to-image generation, where interpretable concept manipulation (cf. Liu et al. 2023b on concept neurons in diffusion models) remains an active area of interest.

**Broader Impact Concerns:**

No additional broader impact statement is required. The authors already provide a thoughtful discussion on bias mitigation in Section 6 ("Addressing Bias"), demonstrating how the framework's interpretable concept space enables direct probing and correction of learned biases - such as gender associations - through adjustment of ground-truth entailment probabilities.

**Claims And Evidence:**

No

**Claims Explanation:**

## Claims

Core claims of the paper (in Section 1 and 6)
- [CL-1]: Efficient multimodal learning
- [CL-2]: Modular learning and adaptation

[CL-S]: Softer claims made through the paper: lower footprint, no fine tuning, interpretability

## Core Issues (CI)

### [CI-1] Ablation studies (Figures 2-3) do not convincingly support efficient multimodal learning claims

The ablation comparing pretrained vs. fresh concept spaces is presented as central evidence for CL1, but has several limitations:
- (a) Convergence benefit may reflect initialization rather than conceptual efficiency. The learning curves for COCO start around 0.4 accuracy, which is unsurprising given that attributes in COCO are highly correlated (e.g., "jumping, catching, exercising" co-occur frequently, as shown in Fig. 1 of Patterson & Hays 2016). Random chance baselines achieve high average precision for frequent attributes (Fig. 6, Patterson & Hays 2016), and the COCO Attributes paper explicitly discusses "benefits of exploiting multi-label occurrence.". The paper reports average thresholded F1 scores. Being able to directly compare AP/mAP scores would help assess whether the observed benefit comes from the concept space structure or simply from exploiting attribute correlations.
- (b) This is not a multimodal alignment task. The ablation tests single-modality projection to attributes, not cross-modal efficiency, yet is used to support claims about multimodal learning.
- (c) No external efficiency comparison. The ablation is entirely internal (pretrained concept space vs. fresh concept space). There is no direct comparison of training footprint, convergence speed, or data efficiency against baseline multimodal methods like CLIP or BLIP.

### [CI-2] Evidence for modular adaptation (CL2) is limited

The claim that "projection models can align with existing conceptual representations rather than learning from scratch" and that the framework enables "effortless incorporation of new modalities" is not empirically validated:
- No experiments demonstrate assimilation of genuinely new concepts or modalities to an existing system
- Each projection model is individually finetuned per dataset/task (COCO, CLEVR, GQA separately), which undermines the modularity claim

### [CI-3] Contradiction regarding finetuning

Section 4.2 states the model is "not finetuned" (Table 2), but Appendix D describes joint training. This inconsistency needs clarification - it appears the method does involve joint training on the image-text matching task? If so should be acknowledged claims about fine-tuning in Section 1 "Contribution".

### [CI-4] Limited experimental scope insufficient for broad claims

- Experiments use small-scale, controlled settings: CLEVR (synthetic), simplified COCO and GQA subsets (only 64 concepts)
- The natural language modality consists of synthetically generated, structured descriptions (essentially concatenated attribute lists), making it difficult to assess true cross-modal alignment capabilities vs natural language from a truly new text modality


## What is well supported

The paper does provide convincing evidence for softer claims (SC):
- **Interpretability:** The ability to probe entailment probabilities (Table 4, Figure 4) is genuinely useful and well-demonstrated
- **Structured reasoning:** The VQA experiments (Section 4.3) show that purely symbolic programs can operate over learned concept representations with a high accuracy
- **Competitive performance:** Tables 1-3 show the framework achieves reasonable performance with limited task-specific optimization

**Requested Changes:**

## [RC]: Requested Critical Changes (Required for Acceptance)

### RC0: Reposition claims

Suggestion is to either reposition claims to focus on interpretability and structured reasoning benefits (which are well-demonstrated), or proceed with RC1 onwards below.

### RC1: Address multimodal efficiency [CI-1]

Address concerns in [CI-1]. Or instead focus on the clearly multimodal image-text matching task via joint fine-tuning (Section 3.3). There appear to be promising results in Appendix D "joint training converges in 600 steps with significantly less resources". Expanding on details with a direct comparison vs (BLIP, VILT, CLIP) on training footprint, or efficiency of fine-tuning would make the claim more convincing.

### RC2:  Empirically validate modular adaptation claims [CI-2]

Show actual modality/concept acquisition: Conduct experiments demonstrating addition of new concepts or modalities to an existing system. For example:
- Modality extension: Test robustness to perturbed text descriptions (language translations, different phrasings) to show decoupled projections are more robust than end-to-end finetuned models in image-text matching
- Concept extension: Experiment with held-out concepts (e.g, CLEVR-CoGenT experiment from NS-VQA Yi 2019)
- Explain per-dataset finetuning: Projection models appear to be individually finetuned per dataset and task. Demonstrate joint training of shared projection models across all datasets (COCO, CLEVR, GQA) with individually trained concept spaces OR explicitly discuss why individual finetuning is necessary and how this impacts the modularity claim.

### RC3: Address fine-tuning contradiction [CI-3]
Refer above for Section 4.2 and Table 2

## [RS]: Requested Strengthening Changes (Non-critical, improve submission)

### RS1: Clarity of presentation and missing details

1. Section 4: Cite COCO attributes (Patterson, 2016) and consider referring readers to Fig 1 from the paper to make it clearer to readers unfamiliar with notion of categories vs attributes in such datasets/tasks.
2. Section 4 / Appendix B: appears like modified subsets of COCO and GQA were used with only 35 attributes/64 concepts (out of 196) for COCO and 64 concepts (our of 620 attributes) for GQA. Why was this necessary? This is an important detail for readers to assess results and should be called out in main contents. Exact attributes/concepts should also be listed in appendix so results are reproducible and comparable with other published results (Patterson, 2016).
3. Section 3.1: For benefit of the reader unfamiliar with this technique when you "adopt .. by Li et al", name it explicitly "probabilistic box embeddings" and provide geometric intuition i.e. overlap of d-dimensional boxes.
4. Section 3.1: Explain the motivation for negative concept sampling: "why are negative concepts/noise added to entailment probabilities? Is it to address noise/sparsity similar to n-gram smoothing?"
5. Section 3.2: Refer readers to Section 4 for attribute vs category distinction and why treated differently in loss
6. Section 4.1: Consider using threshold independent mAP instead of F1, for comparison against (Patterson, 2016) and reproducibility without needing to provide attribute-specific thresholds. Ideally at least one external baseline comparison to establish reasonable attribute prediction.
7. Section 4.1: How does one "fit" a concept space to 15 concepts in CLEVR using batch size 256 when there are only 15C2=105 pairs? This should be explained coherently in one place with cross-references. This should be explained clearly in one place with minimal cross references (Section 3.1, 4.1, Appendix A)
7. Section 4.2 / Appendix D: It seems like baselines were finetuned only on sentence swapping and not attribute swapping? What about the joint trained results from the paper?
8. Section 4.3: Specify whether you used provided pretrained models from IEP (Johnson 2017b) or trained your own. If training your own, explain why this was necessary and provide sufficient details for reproduction
9. Section 4.3: Are all question types from CLEVR evaluated with the functional programs? Explain how spatial comparisons (e.g., "in front" / "behind") are handled using only 2D object detection coordinates. Reference to Fig. 2 from Johnson 2017b might be helpful.


### RS2: Structural improvement

1. To provide room for technical details from appendices, consider shortening redundant claims and information in Sections 1, 6, 7 and moving the ablation in Section 5 to appendix.
2. Consider moving Fig 7 and Fig 8 to main contents as they are critical to understand the application of the framework. Fig 6 also seems relatively redundant w.r.t. Fig 8

### RS3: Broader context and discussion

1. Cite relevant attribute datasets: Section 6 calls for attribute datasets but should cite: VAW (Pham 2021), LSA/TAP (Pham 2022), and OVAD (Bravo 2023). Note that OVAD also evaluates zero-shot attribute prediction of VLM foundation models, which is relevant to this work.
2. Relationship to VLM paradigm: Directly address whether/how this approach relates to or could complement current vision-language models. Specifically:
    - What would it take to handle more general scene understanding with complex natural language queries (as in full GQA) as opposed to controlled symbolic reasoning (CLEVR)?
    - Could the concept-space approach be integrated with or benefit from expansive knowledge in pretrained vision-language models?
    - Natural language as concept space: How do you address the perspective that natural language itself already serves as a generalizable concept space (as leveraged by models like CLIP)?

---

> ### Author Response · Authors · 2025-11-19
> **Response to Reviewer KSiN**
>
> We sincerely appreciate reviewer KSiN for their constructive suggestions and clear feedback on our work, and we are encouraged that they found our framework to have strong motivation and an interesting technical approach. Please find our changes and responses below.
>
> # Changes made to the paper
>
> ## Requested Critical Changes
>
> - RC0:
>
>     We appreicate the two constructive directions the reviewer has provided and we choose to address RC1 and onwards.
>
> - RC1 Address multimodal efficiency:
>     - Added new dataset
>
>         We have included a new unified dataset that combines all concepts and representations from CLEVR, GQA, and COCO. This unified dataset introduces a more challenging and informative setting, where overlapping concepts appear across different data domains and the framework must link representations from these different domains to the *same* underlying concepts. It also contains a substantially larger set of 130 concepts. A detailed description of this new dataset is provided in Section C1 and Table 10 on page 22 & 23 of the revised paper.
>
>     - Reported mAP
>
>         We have reported new mAP measures in both Table 1 and Figure 2 of the revised paper.
>
>     - Cross-modality alignment and efficiency
>
>         We appreciate the reviewer’s suggestions on expanding the promising results in Appendix D, and we have designed a new experiment to evaluate the cross-modality alignment of projection models without fine-tuning, as well as the joint training efficiency compared to external methods including FLAVA, ViLT, and CLIP. (Due to extremely limited GPU resources currently available to us, we could not fit BLIP into our GPU because of its large memory requirements.) This new experiment is reported as a new section (Sec. 4.2) in our revised paper.
>
>         Specifically, we first evaluate the alignment between our vision and natural language projection models after they are trained independently. We use accuracy on the image–text matching task as a proxy to measure the degree of cross-modality alignment. The results in Table 2 demonstrate that our decoupled projection models display strong alignment with each other even though their training is entirely independent, thanks to the shared knowledge space they are adapted to during training.
>
>         Then, only for the purpose of illustrating cross-modality alignment efficiency, we apply the optional joint-training objective to finetune the two projection models and compare their behavior with benchmark multimodal models in Figure 3. The results show that our projection models require significantly less GPU time during the finetuning stage, reaffirming the findings from the earlier results reported in Appendix D.
>
> - RC2 Empirically validate modular adaptation claims:
>
>     - Modality Extension
>
>         We agree with the reviewer’s suggestion on modality acquisition and have designed another new experiment to demonstrate the scalability of the proposed framework through the addition of three new modalities. This experiment is included as a new section (Sec. 4.3) in our revised paper. Specifically, the natural language descriptions previously used for the English modality are translated into Chinese, French, and Spanish to create three new modalities on top of the two existing modalities, vision and English (previously referred to as natural language).
>
>         All five projection models are trained independently with no joint training. As before, we use a cross-modality representation matching task as a proxy to evaluate their alignment in the shared knowledge space. The results in Table 3 show that the newly added modalities exhibit strong alignment with the existing modalities as well as with each other. To gain further insight into their agreement, we report the entailment probabilities of their projections for positive and negative representation pairs in Figure 4.
>
>         This new experiment demonstrates that adding a new modality into the system only requires training a new projection model to adapt to the universal knowledge space. No modification or fine-tuning of the existing projection models is required, which stands in contrast with many mainstream multimodal models.
>
>     - Per dataset finetuning
>
>        We appreciate the reviewer’s question regarding why the projection models are trained separately for each dataset, which motivated us to design the unified dataset described above. This new dataset has been incorporated into Table 1 and Figure 2, and all new experiments in Sections 4.2 and 4.3 are conducted using this larger and more challenging dataset.

---

> > ### Author Response · Authors · 2025-11-19
> > **Response to Reviewer KSiN (cont’d)**
> >
> > ## Requested Critical Changes (cont’d)
> > - RC3 Address fine-tuning contradiction:
> >
> >     We agree with the reviewer’s concern about the potential confusion regarding whether fine-tuning is used. We have moved the cross-modality joint training section from the main body of the paper to the appendix and have clearly indicated that this joint training stage is optional. We also added explicit clarifications in Sections 4.2 and 4.3, both of which involve cross-modality evaluation, to specify exactly when joint training or fine-tuning is applied. All remaining results, including those from the VQA task, are produced by models that do not undergo any fine-tuning. In particular, for VQA, once the program generator is pretrained, it remains frozen during evaluation, together with the frozen concept space and the frozen vision projection model. None of these components are jointly finetuned, in contrast to the VQA baselines, which employ end-to-end optimization.
> >
> > ## Requested Strengthening Changes
> >
> > - RS1 Clarity of presentation and missing details:
> >
> >     1. We have added a citation to COCO Attributes in the experiment section where we introduce the datasets on page 6, and we have also added a reference to Figure 1 from that paper in the dataset details section on page 23.
> >     2. We have added specific details for these datasets in Tables 8, 9, and 10 in the appendix of the revised paper.
> >     3. We have added an explicit description of the probabilistic box embedding space and explained how it produces entailment probabilities by linking the computation to its geometric intuition in Sec. 3.1 on page 4.
> >     4. We have added an explanation of why negative sampling is required in the second paragraph on page 5. In short, the training samples from the real datasets contain only real representations with valid concept overlaps. Without negative sampling, the model would never observe negative concept pairs during training, and the concept space would not learn to distinguish non-overlapping concepts.
> >     5. We have added a reference to Section 4 and have also slightly restructured Section 3.2 to improve the overall flow.
> >     6. We appreciate the suggestion to use mAP and have updated all metrics that previously reported F1 scores to instead report mAP.
> >     7. We agree with the reviewer that if concept pairs for training the concept space were sampled from the full set of concepts, the mathematical formulation would not hold. We would like to clarify that this is not how the concept space is trained. Instead, we use real samples from the dataset, each of which comes with a ground-truth set of concepts derived from its modality-specific representation (for example, an image). For every sample, we enumerate all valid concept pairs within its labeled concept set and use those pairs to supervise the concept space. This ensures that the training signal reflects true concept co-occurrence patterns as observed in datasets. We have also expanded the explanation of this process on page 5 of the revised paper to reduce the need for cross-referencing.
> >     8. The reviewer is correct that the baselines were fine tuned only on the sentence swapping task, and we agree that including joint sentence and attribute training would also be valuable. However, due to the limited rebuttal time frame and GPU resources, we prioritized the new experiments requested by the reviewers. We would be more than happy to add this additional baseline in the final version.
> >     9. We have added an explicit description of how the Program Generator is trained on page 11. We did not use the pretrained model provided in the original work. Instead, we followed the same pretraining setup described in that work to obtain the model used in our experiments.
> >     10. We appreciate the reviewer’s suggestions and have added an explanation of how spatial information is obtained on page 12 in the Object Detection and Projection section. And all question types appeared in our evaluation.
> >
> > - RS2 Structural Improvement
> >
> >     We agree with the reviewer’s suggestion and have moved the ablation study to Section A.4 in the appendix.
> >
> > - RS3 Broader context and discussion
> >
> >     1. We find the attribute datasets pointed out by the reviewer to be closely related to our work, and we have added citations to these datasets in the Call for Concept-Focused Datasets section under Sec. 5 on page 13.
> >     2. We appreciate the reviewer’s suggestion to discuss potential ways to connect our method with more widely adopted vision–language foundation model architectures. We are excited to share that we are actively investigating approaches in this direction. We have added a new discussion segment on page 13 regarding this topic and would like to refer the reviewer to the revised paper for a more detailed description.

---

> > > ### Author Response · Authors · 2025-11-19
> > > **Response to Reviewer KSiN (cont’d – 2)**
> > >
> > > We sincerely thank the reviewer again for their time, careful reading, and constructive feedback. We hope our revisions and clarifications have fully addressed all raised concerns, and we are very happy to make any additional changes that would further improve the clarity, completeness, or quality of the paper.

---

> > > > ### Comment · Reviewer_KSiN · 2025-12-04
> > > > **Response to author's comments & revision**
> > > >
> > > > Appreciate the openness from the authors in addressing all suggestions (both critical and strengthening) in detail, and their time to make significant revisions and experiments.
> > > >
> > > > Overall the revised version greatly improves comprehensibility of the paper and convincingness of claims, and I have updated my recommendation accordingly. Wish the authors the best.
> > > >
> > > > (miscellaneous notes included below to loosely consider for a final pass)
> > > >
> > > > Misc notes:
> > > > - Interesting to see learning curve gap between regular projection head and concept based increase as the task gets more difficult in Fig 2 (e.g. very clear for the unified task). Consider calling out in caption as it makes the case
> > > > - Explicitly mention the 130 size concepts when introducing Unified exp in section 4. Also do a pass to make sure it is clear to reader in later sections whether unified/per-dataset projection models are used in following experiments
> > > > - Unresolved/invalid “as indicated in Fig” reference in Appendix B
> > > > - w.r.t. Section 4.3 worth noting that in practice one would likely use LoRa adapters vs maintaining entirely separate BERT-like models as a modular compact way to make controllable extensions to modalities/concepts.

---

> > > > > ### Author Response · Authors · 2026-01-20
> > > > > **Response to Reviewer KSiN**
> > > > >
> > > > > We would like to thank reviewer KSiN for their continued time, effort, and constructive engagement in helping us improve this paper.
> > > > >
> > > > > We have addressed the noted clarity issues regarding the Unified dataset in the main text, clarified the relevant experimental descriptions, and corrected the unresolved figure reference. We also appreciate the reviewer’s insightful discussion on the use of LoRA adapters as a practical and modular approach for extending model support to additional languages
> > > > >
> > > > > Thank you again for your thoughtful feedback and encouragement throughout the review process.

---

### Review · Reviewer_jDaK · 2025-11-06

**Summary Of Contributions:**

The paper proposes a modality-agnostic concept space designed to capture structured, abstract knowledge. This is accompanied by modality-specific projection models (ViT, ResNet) that map raw inputs from different modalities onto this single shared space.
The paper claims the shared embedding space is modality agnostic and that this approach allows the shared concept space to possess abstract knowledge, which in turn facilitates more efficient learning and effortless incorporation of new modalities. The authors also claim that their framework, with a modest pretraining footprint, achieves comparable performance to benchmarks out-of-the-box without fine-tuning.

The authors first describe their approach using a mathematical basis of entailment probabilities, then describe the parameterization of the concept space as box probabilities. The authors then detail how modality projections could be learned by fitting to the bounds of the box probabilities using standard MLP + relu blocks. They then demonstrate the performance of their approach on downstream tasks, specifically image-text matching and VQA on CLEVR, GQA and COCO datasets, and compare it to existing competitive models to showcase the effectiveness of their framework, while highlighting the efficacy of adaptation and ease of interpretability in their approach.

Strengths:
- Shows a learned concept space that is interpretable and specifies projections into this space demonstrating the efficacy of introducing new modalities
- Substantiates their claims through multiple ablations and downstream tasks on text-matching and VQA, specifically noting that beating state of the art is not their emphasis, rather it's the interpretability provided by a concept learning approach.

Weakness (See more below):
- Does not provide sufficient evidence around **cross modal** entailment/interpretability
- Baselines for learning the shared space seem weak, with not enough ablations to showcase the efficacy of their chosen embedding space compared to any learned embedding space of similar size.

**Additional Comments:**

Thank you for your submission to TMLR, I look forward to hearing your response.

**Audience:**

Yes

**Audience Explanation:**

Yes, the concept-centric approach and interpretability arguments can be topics of interest to TMLR audience.

**Broader Impact Concerns:**

There is no Broader Impact statement and is not required.

**Claims And Evidence:**

No

**Claims Explanation:**

Several claims in the paper are not fully supported by the evidence provided or require clarification.
1. On Interpretability: The paper claims that inference is "performed entirely within a shared space of learned concepts that offers interpretability." The evidence for this ("Probing this concept space can also be achieved through simple queries of concept pairs...") is limited. The paper positions interpretability as a key advantage of their approach, but most of the evidence is moved to the Appendix section. The interpretability demonstrated is only shown along pairwise relationships on text (specifically words and short tokens). This does not sufficiently back the broader claim of multimodal interpretability. I would recommend adding visualizations (T-sine or other) of the learned concept space, or showcase vision-text entailments to strengthen this claim.

2. On Efficient Learning and Adaptation: The paper claims "more efficient learning curves" and adaptation to new modalities in "a few 100 steps."
- This "faster learning" argument is a bit weak. The experiment compares the proposed method (which uses a pretrained embedding space) to a baseline MLP - that likely does not have access to the pretrained space. A model leveraging a pretrained space will naturally learn faster. Consider adding a few more ablations around models that have access to pretrained embedding spaces.
- The claim of adapting in a few hundred steps also seems tied to the dimension D of the box embedding space, which is not discussed. The output dimension of the MLP is not specified, so there’s no way to accurately compare the parameterization of the learned concept embedding space, v/s any learned latent space.
- The claim of efficiency via "referencing a (compact parameterized) concept space" is not supported by evidence. What is specifically efficient about the concept space v/s other learned spaces of a similar dimension?

3. On Novelty and Related Work: The authors claim that "this idea of a concept-focused learning scheme has rarely been explored in the field of multimodality learning."
- Sharcs (https://arxiv.org/pdf/2307.00316) appears to be a very close piece of related work, where shared concepts are learned and mapped to a semantically meaningful space.
- Unified-IO Line of work (https://arxiv.org/abs/2312.17172 ) also explores unified representations using patch sequences.

4. On Empirical Evidence:
- The CLEVR dataset and it’s associated metrics seem to be very saturated (upper 0.99 accuracy), limiting the conclusions that can be drawn. While I appreciate the call to action for more datasets for concept learning, the authors should consider more recent datasets (e.g CURI - https://arxiv.org/abs/2010.02855 )
- No confidence intervals (CIs) are reported in Table 2, making it hard to know if the reported numbers are statistically significant or if the vision encoders are doing the heavy lifting in the metrics.
- In Table 1, a regular MLP baseline actually performs better on some tasks on CLEVR than the proposed method, undermining the claim of being "competitive." Consider a few more experiments other than a regular MLP to provide a range of baselines in learning the concept space for Table 1.
- The paper does not demonstrate how discrete tokens (text) can be blended with continuous tokens (image patches). This is a critical challenge in unified projection. The authors mention projecting vision and text embeddings into a continuous probabilistic space but do not comment on the efficacy of projecting continuous tokens. While the box diagrams shown in Figure 7, are helpful, consider also showing vision-vision and text-text entailments as a way to demonstrate modality agnostic relationships learning in the concept space.

5. Misc Unsupported Claims:
There's a minor claim in the paper that states in Section 3.3 on the training loss - "Optionally, the optimization can also include parameters from C, so that the abstract knowledge learned in the concept space is adjusted based on modality-specific information." There is no experiment to support this claim. Consider specifying this in future work or adding experimental evidence.

6. On the Motivational Analogy: The paper uses the analogy of learning a new language, where "we intuitively connect new linguistic elements to our existing understanding of the world." This analogy is flawed; learning a new language is still (initially) within the text modality. Humans sometimes translate first, then build new concepts over time, which is a different process. While the comparison is noted, it weakens the argument. Consider another comparison, particularly around learning concepts from different modalities, since that seems to be one of the underlying claims of the paper.

**Requested Changes:**

1. Clarifications:
- Provide a clear justification for the mathematical formulation of $P(y)$ as a product of $m()$ over all dimensions for the d-dimensional embedding K. Why does this need to be a product of probabilities *across* the dimension, rather than leave it as is or a summation? How does this change the outcome of the experiments as the dimension D varies?
- **Clarify what is novel in the multimodal projection and embedding space, given that the concept space itself is adopted from Li et al (2018) (Section 3.1 in the paper)**. You can explicitly call out the changes made to adapt it to a multimodal concept space.
- State the output dimension of the baseline MLP used in the experiments.
2. Experiments and Evidence:
- Interpretability: **The paper should show evidence of multimodal relationships in the concept space, not just pairwise text relationships.**
- Projections: Consider a demonstration and/or discuss how discrete tokens (text) are blended with continuous tokens (image patches). Comment on the efficacy of the continuous token projections, where both vision and text embeddings are split to learn parameters of the box embedding.
- Baselines:
  - **The "faster learning" claim must be substantiated with a fair comparison**. The baseline MLP should also have access to a pretrained embedding space that serves as its projection.
  - Consider adding other baselines that use the same vision and language backbones. The BEIT line of work (e.g., https://arxiv.org/abs/2208.10442) could serve as a baseline, as it also learns a common embedding space and uses the same vision and text encoders.
- Metrics: Consider showing the F-1 scores and other metrics mentioned in Table 1 for the training accuracy comparison in Figure 3. It can strengthen the claim that the concept space enables faster learning.
- Ablations: Consider adding an ablation study on the dimension of the concept space embedding (K) to justify the choice of 50.

3. Scope and Limitations:
The paper suggests projections along "attributes" and "categories." Many modalities (e.g., robotic actions, alpha channels) do not fit this construction. Consider explicitly calling out this limitation, it is fine to leave this as future work.

---

> ### Author Response · Authors · 2025-11-19
> **Response to Reviewer jDaK**
>
> We sincerely appreciate reviewer jDaK for their constructive feedback on our work. Please find our changes and responses below.
>
> ## Changes made to the paper
> - **On Novelty and Related Work**
>     We thank the reviewer for pointing us to these relevant works. SHARCS, in particular, is highly aligned with the theme of concept-based multimodal learning, and we have now incorporated it into our related-work discussion along with the Unified-IO line of work. We also adjusted our wording from “rarely explored” to “under-explored” to better reflect the broader landscape. SHARCS learns concepts through modality-specific encoders under CBM-style supervision and then aligns those modality-dependent concept representations. In contrast, our framework first learns an abstract concept space independently of any modality and subsequently adapts each modality to that shared space, resulting in a decoupled training scheme. The Unified-IO family follows a unified modeling direction where multimodal inputs are transformed and fused within a single architecture, which also differs from our separation between learning concept space and learning projection models.
>
> - **Optimization including concept space parameter**
>     We agree with the reviewer and have removed that description from the main body of the paper.
>
> - **Clarifications**
>     - On the effect of concept dimension $K$ on experimental outcomes: we have added an ablation study to evaluate this effect and refer the reviewer to the ablation section below for details.
>     - On the dimension of the MLP: the MLP head used as a baseline takes the same backbone embeddings (from ViT, BERT, and ResNet) as input and outputs a vector whose dimension equals the total number of concepts in the dataset. We added a detailed description of its output shape and layer dimensions in Section D.2 on page 24 of the revised paper.
>
> - **Added mAP as a metric**
>     We have added mAP as an evaluation metric in both Table 1 and Figure 2.
>
> - **Ablation on Concept Space Dimension**
>     We have added a new ablation in Section A.5 on page 21, where we evaluate concept space dimensions $K \in \{24, 50, 96\}$. The results in Table 7 show that $K = 50$ provides the most consistent performance across datasets. A reference to this ablation is also added at line 4 on page 7.
>
> - **Scope and Limitations**
>     We appreciate the reviewer’s suggestion and have added a Scope and Limitations subsection discussing the limitations of the framework for certain modalities. This appears in the Discussion section on page 14.
>
> - **New unified dataset**
>     We appreciate the reviewer’s suggestion to consider CURI for evaluation. We have added a citation to CURI in the discussion of concept datasets on page 13. Since our experiments already include CLEVR, which occupies a data domain very similar to CURI, we chose not to add CURI directly. Instead, we constructed a new unified dataset combining all concepts and representations from CLEVR, GQA, and COCO. This dataset introduces a more challenging and informative setting, where overlapping concepts appear across different data domains and the framework must link representations from these different domains to the same underlying concepts. It also contains a substantially larger set of 130 concepts. A detailed description is provided in Section C1 and Table 10 on page 23.
>
> - **Additional experiment to showcase vision–text entailment**
>     We added a new experiment in Section 4.2 demonstrating the cross-modality alignment of our independently trained vision and text projection models. These models achieve strong alignment/entailment in the concept space without any joint fine-tuning, which stands in contrast to mainstream multimodal methods that rely on explicit joint training.
>
> - **Other changes**
>     - We invite the reviewer to read the new Section 4.3, where we demonstrate the scalability of the framework by incorporating three new modalities without modifying existing projection models.
>     - We changed the word “competitive” to “comparable” in the abstract to more accurately reflect our experimental results.

---

> > ### Author Response · Authors · 2025-11-19
> > **Response to Reviewer jDaK (cont’d)**
> >
> > ## Clarifications
> >
> > Now, we would like to offer some clarifications regarding the reivwer's questions.
> >
> > - On Concept Space
> >     - We agree with the reviewer that the concept space used in our framework is adopted from Li et al. (2018). While this concept space is an important building block of our approach, we do not position it as our main contribution. The original work by Li et al. focuses on learning hierarchies, partial orders, and lattice structures in a purely abstract setting, with experiments on datasets such as WordNet and MovieLens. In contrast, our framework introduces a projection scheme that allows the knowledge encoded in this concept space to be applied to real modalities such as images and natural language, rather than being restricted to abstract symbolic structures. This extension enables the concept space to support multimodal reasoning and alignment in a way that was not explored in the original formulation.
> >     - Consequently, we did not include the full mathematical details or derivations of the formulas for $P(\cdot)$ and $m(\cdot)$. In Section A1 & A2 on page 19 of the paper, we provide a concise description of these equations together with the core intuitions behind the concept space, and we refer readers to the original paper for further technical details.
> >     - In this probabilistic box embedding space, each concept is a d-dimensional latent box, and entailment is modeled through the intersection volume between two boxes. Since the volume of a box is the product of its side lengths along each latent dimension, the entailment probability must also be a product across dimensions.
> > - On Interpretability
> >     - We wanted to offer some clarifications on the entities in concept space and our demonstration of the interpretability in Table 6 at page 13 and Figure 5 at page 20 of our paper. All the concept entities in the concept space correspond to abstract semantic units such as red, metal, and car. These concepts are modality agnostic, and they do not come from modality specific representations. The words shown in the tables and figures are simply human readable names for these abstract concepts rather than text derived features/projections.
> >     - As the concept space is a geometrical probabilistic box embedding space that captures entailment relations through the "volumn" of overlap between boxes, we believe that directly presenting the learned entailment probabilities offers a more faithful and informative demonstration of interpretability than visualizing the boxes through dimensionality reduction tools such as t-SNE.
> > - On Baseline Comparison
> >     - We would like to clarify that the baseline MLP method uses the same pretrained backbones as our projection models, and both the MLP head and our projection head are initialized with random weights. This setup ensures that the comparison focuses specifically on whether modality-agnostic abstract knowledge in the concept space helps the learning process of modality-specific projection models. As noted in the last line on page 1, access to a shared knowledge space is an intended property of our framework, reflecting how humans learn new modalities by leveraging prior knowledge rather than starting from scratch.
> > - On Efficient Learning of Concept Space
> >     - We would like to direct the reviewer to Sec. 4.1 (page 7), where we note that the concept space contains only a few thousand parameters for a moderately sized configuration. In our experiments, training this concept space to convergence requires only tens of minutes.
> > - On Handling of Discrete vs. Continuous Tokens
> >     - We appreciate the reviewer’s question regarding how discrete text tokens and continuous image patches are handled within our framework. We would like to clarify that our method does not operate on individual tokens or patches. Instead, as stated in Section 4.1 (page 7), both the ViT-based and BERT-based projection models use their backbone’s pooled [CLS] embedding as the modality-specific feature representation. This embedding is already a continuous vector regardless of whether the input originated from discrete tokens (text) or continuous patches (vision). Therefore, the modality-specific projection head in our framework receives a single continuous representation embedding vector of its modality input, and projecting such vectors into the probabilistic concept space is straightforward (with just one linear layer) and decoupled with other modalities. <br> The efficacy of handling continuous vs discrete tokens is therefore determined by the choice of backbone encoder (for example, ViT, BERT, or ResNet), not by the design of our framework. A key advantage of our approach is precisely this decoupling: any modality, whether text, image, or others, can be integrated as long as a backbone provides a fixed-dimensional continuous embedding to be used as input for the final projection head.

---

> > > ### Author Response · Authors · 2025-11-19
> > > **Response to Reviewer jDaK (cont’d – 2)**
> > >
> > > We sincerely thank the reviewer again for their time, careful reading, and constructive feedback. We hope our revisions and clarifications have fully addressed all raised concerns, and we are very happy to make any additional changes that would further improve the clarity, completeness, or quality of the paper.

---

> > > > ### Comment · Reviewer_jDaK · 2025-12-07
> > > >
> > > > Thank you for your updates and clarifications, I have updated my recommendation accordingly. The new sections and experiments improve the reading greatly and provide clarity on the claims. All the best to the authors!
> > > >
> > > > Couple of minor notes to further improve clarity for your consideration:
> > > >
> > > > 1. Clarify the definition of new modality in Section 4.3, I was expecting a modality like audio etc, but I can understand that would have taken an outsized amount of work to showcase. Maybe consider rewording the details a bit to emphasize why the newer languages are considered as completely different modality spaces inline with how the definition of modality is treated in other parts of the paper - i.e how having modality specific projections for each language would be different than using a BERT trained on those new languages in the existing text modality.
> > > > 2. Consider briefly mentioning the size of the datasets for the new languages. If these required few samples/training steps, that further showcases the advantage of your approach.
> > > > 3. The evidence in Figure 3 is especially strong, maybe mention that the Validation Accuracy is from image-text matching from the Unified dataset in the caption. Also mention the dataset for the accuracy in Table 2 (validation or test) to improve clarity.

---

> > > > > ### Author Response · Authors · 2026-01-20
> > > > > **Response to Reviewer jDaK**
> > > > >
> > > > > We would like to thank reviewer jDaK for their continued time and effort in helping us improve our paper.
> > > > >
> > > > > We have addressed the comments in the second and third paragraphs of Section 4.3 and the caption of Figure 3.
> > > > >
> > > > > Thank you again for your thoughtful feedback and constructive engagement throughout the review process.

---

### Author Response · Authors · 2025-11-19
**Summary of Revisions**

We would like to sincerely thank all reviewers and the action editor for their time, effort, and thoughtful feedback, which greatly helped us improve the paper. We are encouraged by the assessment that our proposed framework is of interest to the TMLR audience (reviewers jDaK, KSiN, and 5zbu), and we appreciate the recognition that our motivation is strong, well positioned, and promising (reviewers KSiN and 5zbu). We are also grateful for the reviewers’ remarks that the proposed method is interesting (reviewers KSiN and 5zbu) and we appreciate the reviewers’ recognition of the interpretability afforded by the learned shared concept space (reviewers jDaK and KSiN).

We would now like to offer a brief summary of the revisions we have made to address the reviewers’ valuable suggestions.

1. Creation and addition of a new unified dataset to evaluate generalizability.

    We have created and added a new unified dataset that merges all concepts and representations from CLEVR, GQA, and COCO into our evaluation. This dataset contains 130 total concepts and provides a more challenging and informative setting for assessing generalization. Several concepts appear across multiple datasets but with different visual or textual realizations (for example, synthetic CLEVR scenes versus real-world GQA and COCO images referring to the same concept). As a result, the framework must learn to associate heterogeneous representations from different data domains with the same underlying abstract concept, offering a more realistic and demanding test of generalizability. A detailed description of the dataset is provided in Section C1 and Table 10 of the revised paper.
2. Addition of a new experiment on cross-modality alignment and efficiency.

    We have added a new experiment in Section 4.2 to more fully evaluate cross-modality alignment and training efficiency. The experiment extends the promising results previously shown in Appendix D by examining how the vision and natural language projection models align with each other when trained independently, as well as how efficiently they can be jointly finetuned compared with mainstream multimodal models such as FLAVA, ViLT, and CLIP. (Due to limited GPU resources, BLIP could not be included because of its memory requirements.)

    The first part of the experiment measures alignment between independently trained projection models using an image–text matching task. Results in Table 2 show that the projection models achieve strong cross-modality alignment even without any joint training, benefiting from the shared knowledge encoded in the concept space.

    The second part evaluates cross-modality alignment efficiency. For illustration purposes, we apply the optional joint-training objective to finetune the two projection models and compare their behavior to external baselines. Figure 3 shows that the projection models require substantially less GPU time during finetuning, consistent with the earlier observations reported in Appendix D.

3. Addition of new experiment on modality extension and generalization

    We added a new experiment in Section 4.3 to evaluate how well the framework scales to new modalities. We created three additional language modalities (Chinese, French, and Spanish) by translating the original English descriptions, and we trained their projection models independently alongside the existing vision and English models. Using the same cross-modality matching task as before, Table 3 shows that all five modalities align strongly in the shared concept space without any joint training. Figure 4 further reports the entailment probabilities for positive and negative cross-modality pairs.

    This experiment demonstrates that incorporating a new modality only requires training a corresponding projection model to adapt to the universal knowledge space. No adjustment or fine-tuning of the existing projection models is required, which contrasts with many mainstream multimodal approaches.

4. Additional improvements and clarifications throughout the paper.

    Beyond the new experiments and dataset, we have made several revisions to strengthen clarity and technical completeness. These include adding an ablation study on the effect of concept space dimension, incorporating citations to additional related works, expanding methodological details and clarifications, and improving the overall structure and flow of the paper. We have also updated our evaluation metrics by replacing F1 scores with mAP for clearer comparisons, and we expanded the discussion section to include several forward-looking topics suggested by the reviewers. These changes collectively improve the readability, technical transparency, and completeness of the submission.

---

> ### Author Response · Authors · 2025-11-19
> **Summary of Revisions (cont’d)**
>
> We once again thank all reviewers and the action editor for their thoughtful evaluations and constructive suggestions. Their feedback has led to substantial improvements in the clarity, scope, and completeness of the paper, including new experiments, expanded discussions, and additional technical details. We hope that the revised manuscript addresses all concerns raised, and we are very happy to make further refinements that would improve the paper even more.

---

### Decision · Action_Editor_Ym6g · 2025-12-30

**Recommendation:** Accept as is

**Additional Comments:**

The submission unequivocally meets the bar for the Audience criterion. Opinions are split among reviewers on the Claims and evidence criterion, but after deliberating with the reviewers my conclusion is that while the experiments demonstrate the validity of the proposed approach in a more modest setting, they are sufficient to substantiate the main claims made in the paper.

**Audience:**

Yes

**Audience Explanation:**

All reviewers agree that concept-centric multimodal learning is of interest to TMLR's audience:

* "[The] concept-centric approach and interpretability arguments can be topics of interest to TMLR audience." (jDaK)
* "The paper addresses several questions of significant interest to the multi-modality learning and interpretable AI communities [...]" (KSiN)
* "The idea of concept-centric multimodal learning itself is an interesting direction. Thus, it may still be of interest for some individuals." (5zbu)

**Claims And Evidence:**

Yes

**Claims Explanation:**

Opinions are split among reviewers on the Claims and evidence criterion: Reviewers KSiN and jDaK find that the submission meets the bar, but Reviewer 5zbu disagrees. More specifically:

* Reviewer KSiN noted in their review that the submission's multimodal claims are overstated, and the authors responded with a new experiment that adds three new modalities (Chinese, French, and Spanish). Reviewer jDaK noted in their official recommendation that they were expecting this new Section 4.3 to present a modality like audio or something similar. In discussions Reviewer 5zbu also objected that "we usually will not regard different multilingual types as different modalities". However, Reviewer KSiN felt that this was sufficient to address their concern: "[the proposed approach] is modular by design and the addition of Section 4.3 verifies that independent text projection models trained on different modalities (languages) align with each other in the shared concept space without joint training. This is kind of sufficient, might have been nice to compare with a combined projection model too (as BERT is multilingual). But I don’t think the addition of audio would add much more – especially when image descriptions for these tasks are very simple." Taking a step back, the question here is whether the addition of the three new "modalities" in Section 4.3 adequately substantiates the claim that the abstract knowledge imbued in the proposed approach facilitates "effortless incorporation of new modalities into the framework". While the experiments perhaps do not present the most challenging problem setting, they do demonstrate that the proposed approach makes it possible to incorporate an independently-trained projection model after the fact, and I consider the claim to be sufficiently backed up by empirical evidence.
* Reviewer 5zbu remained concerned that the baselines are relatively old or out-of-date, and that there is no strong baseline to demonstrate the effectiveness of the "cross-modality matching accuracy". Reviewer KSiN pointed out that "[the] authors explicitly call out a non-goal of state-of-art performance against systems tuned for these tasks." Indeed, the submission states: "While our goal is not to surpass existing benchmarks in raw performance, these results support
the viability of a cognitively inspired learning paradigm. Rather than optimizing solely for accuracy, our framework emphasizes learning efficiency, interpretability, and structural alignment with human cognition." Reviewer KSiN also shared that their "larger concern was that [experimental] tasks are very narrow and simple/limited compared to the current VLM paradigm. It does seem like one needs more general decoders over this concept space (also called out by [Reviewer] 5zbu)." They note however that "[this] has been mitigated to some extent by adding future work/discussion on pages 13/14". Taking a step back, I agree with Reviewer KSiN that the experiments are sufficient here.